# Selected Seeds as Sources of Bioactive Compounds with Diverse Biological Activities

**DOI:** 10.3390/nu15010187

**Published:** 2022-12-30

**Authors:** Natalia Sławińska, Beata Olas

**Affiliations:** Department of General Biochemistry, Faculty of Biology and Environmental Protection, University of Lodz, 90-236 Lodz, Poland

**Keywords:** biological activity, active compounds, phytochemicals, phenolic compounds, seeds

## Abstract

Seeds contain a variety of phytochemicals that exhibit a wide range of biological activities. Plant-derived compounds are often investigated for their antioxidant, anti-inflammatory, immunomodulatory, hypoglycemic, anti-hypercholesterolemic, anti-hypertensive, anti-platelet, anti-apoptotic, anti-nociceptive, antibacterial, antiviral, anticancer, hepatoprotective, or neuroprotective properties. In this review, we have described the chemical content and biological activity of seeds from eight selected plant species—blackberry (*Rubus fruticosus* L.), black raspberry (*Rubus coreanus* Miq.), grape (*Vitis vinifera* L.), *Moringa oleifera* Lam., sea buckthorn (*Hippophae rhamnoides* L.), Gac (*Momordica cochinchinensis* Sprenger), hemp (*Cannabis sativa* L.), and sacha inchi (*Plukenetia volubilis* L). This review is based on studies identified in electronic databases, including PubMed, ScienceDirect, and SCOPUS. Numerous preclinical, and some clinical studies have found that extracts, fractions, oil, flour, proteins, polysaccharides, or purified chemical compounds isolated from the seeds of these plants display promising, health-promoting effects, and could be utilized in drug development, or to make nutraceuticals and functional foods. Despite that, many of these properties have been studied only in vitro, and it’s unsure if their effects would be relevant in vivo as well, so there is a need for more animal studies and clinical trials that would help determine if they could be applied in disease prevention or treatment.

## 1. Introduction

Plant-derived bioactive compounds are often investigated for their health-promoting properties [1]. Many currently used pharmacological drugs are either plant-derived or synthetic derivatives of plant-derived substances [2]. Some phytochemicals were observed to have similarly potent effects as their currently used pharmaceutical counterparts while having a superior safety profile and fewer side effects, which makes them potential candidates for drug development [3]. Alternatively, they can be utilized in the production of functional foods and nutraceuticals, which recently garnered a lot of attention due to their potential health benefits [4,5].

Seeds are sources of various beneficial bioactive substances [5,6,7,8,9]. While some plant species are grown for the purpose of seed collection and consumption, in others they are treated as a byproduct and oftentimes discarded—for example during juice or jam production [5,10]. They can be pressed for oil or used to make extracts, which are growing in popularity due to consumers’ rising interest in natural products [10]. The meal (flour) left after oil production still contains numerous bioactive substances and can be used as a food additive as well [5,6]. Moreover, the seeds could be utilized as a source for extraction of various active compounds. Reusing leftover seeds has an additional benefit of minimalizing the waste left after food processing and reducing environmental contamination [11]. In some cases, this process is already underway—for example, extracts from grape seeds are already readily available on the market [12,13]. There are many approaches to extracting different components from seeds. The methods used to extract proteins include alkali extraction, ultrasonic assisted alkali extraction, extraction by fractionation method, enzymatic extraction, or enzyme-assisted micro-fluidization [14,15,16,17]. For example, González Garza et al. [15] used alkaline extraction with isoelectric precipitation to obtain bioactive peptides from *Moringa oleifera* seeds. Polysaccharides can be obtained by hot water extraction, ultrasound-assisted extraction, ultrasound-microwave-assisted extraction, ultra-high pressure-assisted extraction, enzyme-assisted extraction, subcritical water extraction, or pulsed electric field assisted extraction. For example, Tian et al. [16] extracted the polysaccharides from sacha inchi seed powder with the help of hot water reflux method [14,15,16,17].

Assessing average seed intake is not an easy task. Seeds can be eaten on their own, added to various foods like muesli or snack bars, or consumed with the whole fruit. Moreover, as dietary habits are different between different cultures, the contribution of seed consumption to the daily intake of nutrients and phytochemicals may vary significantly between different populations and individuals [18,19]. Most often studies focus only on the intake of seeds overall, and not on the consumption of seeds from individual plant species. Some of the most important nutrients contained in ‘specialty seeds’ (high value and/or uncommon seeds) include polyunsaturated fatty acids, dietary fiber, minerals (like potassium, phosphorus, calcium, and magnesium), vitamins, and amino acids [9]. Unfortunately, seeds can also contain antinutrients that can have negative effects on human health. Saponins, tannins, phytates, lectins, or cyanogenic glycosides can be brought up as an example [9,20]. According to Pojić et al. [21], the main antinutritional factors of hemp seed meal are phytates, glucosinolates, trypsin inhibitors, and condensed tannins. Bueno-Borges et al. [20] have reported, that sacha inchi seeds contained tannins, phytic acid, saponins, lectins, and trypsin inhibitors, though some of these compounds could be almost completely eliminated by thermal processing.

While there are many studies focusing on the association of nuts and seeds consumption with prevalence of different diseases, less is known about the seeds themselves. Studies have linked high intake of nuts and seeds with lower risk of cardiovascular diseases, diabetes, and cancer mortality but they usually include only more popular types of nuts and seeds, like walnut, almond, hazelnut, pecan, pistachio, macadamia, quinoa, sesame, sunflower seed, chia, pumpkin seed, flaxseed, or hemp seed [18,22].

This review describes the phytochemical characteristics and biological properties of seeds from eight selected plant species—blackberry (*Rubus fruticosus* L.), black raspberry (*Rubus coreanus* Miq.), grape (*Vitis vinifera* L.), *Moringa oleifera* Lam., sea buckthorn (*Hippophae rhamnoides* L.), Gac (*Momordica cochinchinensis* Sprenger), hemp (*Cannabis sativa* L.), and sacha inchi (*Plukenetia volubilis* L.). These species represent several different scenarios in which seeds can be consumed—eaten directly with the whole fruit (e.g., blackberry, black raspberry, grape, *M. oleifera*, or sea buckthorn) or separately (e.g., Gac, hemp, and sacha inchi). Among the second group, some plants are cultivated mainly for their seeds which are a part of diet (sacha inchi), while in others, seeds are eaten almost exclusively for their medicinal purposes. Moreover, we selected species that are native to different parts of the world, to provide a wider perspective that includes plants cultivated in different regions. These eight species were chosen to represent different types of chemical compounds found in seeds and illustrate interesting biological activities that have the potential to be utilized in medicine and nutritional science.

This review is based on studies identified in electronic databases, including PubMed, ScienceDirect, and SCOPUS. Last search was run on 23 December 2022. The search terms included the name and alternative names of the researched plant, and the word “seed” or “seeds”. We included both pre-clinical and clinical studies. Subsequent searches further narrowed the results to researched activity or compound—e.g., (ellagic acid) AND ((anti-inflammatory) OR (inflammation)). We excluded articles published more than 20 years ago (counting from the first search in December 2021), articles about plant parts other than seeds, articles about the effects of seeds combined with other plant parts (e.g., skin or pulp), and articles about a combination of extracts or compounds from different plant species.

## 2. Chemical Content of Seeds

Seeds contain many kinds of active substances, categorized by their structure (Figure 1). Polyphenols and their four main categories: flavonoids, phenolic acids, lignans, and stilbenes are among the most well-known phytochemicals [1,9,23]. They are best-known for their antioxidant properties, but the range of their activity is much wider [2,23]. Many pre-clinical and clinical (both observational and intervention) studies have suggested, that increased intake of phenolic compounds is related to a lower risk of cardiovascular diseases, diabetes, cancer, and neurodegenerative diseases [24,25,26,27]. In the past, beneficial effects of polyphenols were attributed mainly to their antioxidant activity, but currently, evidence suggests that dietary intake might not provide a high enough concentration in tissues for radical-scavenging properties to be significant. More research is still needed to ascertain the exact effects, mechanisms of action, and bioavailability of various polyphenols [23]. 

Aside from polyphenols, numerous other types of phytochemicals have been isolated from seeds. They include, but are not limited to phenyl alcohols, phenolic glycosides, phytosterols, terpenoids, terpenophenols, tocochromanols, lignanamides, tannins, nitrogen-containing compounds, and saponins [4,23,28,29,30,31,32,33,34,35,36,37,38,39]. Terpenoids can be divided into several groups, including triterpenoids and tetraterpenoids. Lycopene, a carotenoid present in Gac seeds can be brought up as an example [42,43,44,45,46]. Phytosterols, like campesterol or stigmasterol are natural steroids that can be found in many different plant species. Many studies have noted their positive effect on the cardiovascular system [47]. Tannins are a large group of polyphenols, that can be divided into hydrolyzable tannins and condensed tannins. Hydrolyzable tannins contain a central core of glucose or another polyol that can be esterified with gallic acid (gallotannins) or hexahydroxydiphenic acid (ellagitannins). For example, sanguiin H-6 is a hydrolyzable tannin abundant in blackberry seeds. It shows a wide array of biological activities, including antioxidant, antibacterial, antitumor, anti-angiogenic, estrogenic, and anti-osteoclastogenic [2,48,49,50,51,52]. Condensed tannins are oligomers or polymers of flavan-3-ol linked through an interflavan carbon bond. They are also called proanthocyanidins; examples include procyanidin B_2_ present in grape seeds [53]. Saponins are steroid and triterpenoid glycosides present in many different plant species. They display a wide range of biological properties, both beneficial and detrimental [54]. For example, saponins from Gac seeds had anti-inflammatory activity, but also showed cytotoxic properties [55,56].

Unfortunately, in many cases biological activity of phytochemicals has been studied only in vitro, and the results might not be replicable in vivo. Therefore, there is a need for an increased number of studies that would ascertain if the effects of various substances isolated from plants are relevant in vivo as well [4,57]. Recently, potential pharmaceutical applications of seed proteins and polysaccharides have also been studied [58,59,60,61]. Proteins, protein hydrolysates, and peptides have shown a wide array of biological activities, including antioxidant, antimicrobial, anticancer, or antihypertensive. Bioactive peptides can be obtained from natural sources, like dairy, eggs, fish, nuts, legumes, cereals, fruits, or seeds. They can be formed during external hydrolyzation in the laboratory, in vivo digestion, or food processing. Polysaccharides are also an interesting area of research. They have been investigated for their antioxidant, immunomodulatory, antitumor, anticoagulant, and anticancer properties. Moreover, they can be modified by various methods (e.g., acetylation, selenization, or phosphorylation) to improve their effectiveness. Proteins and polysaccharides have shown various promising biological effects in animal models, but there is a need for more clinical trials that would allow them to be introduced as functional foods, nutraceuticals, and pharmaceuticals [58,59,60,61,62,63,64].

The quantity of substances contained in the seeds depends on the genotype, growing conditions (temperature, exposition to sunlight, water and nutrient availability, or soil pH), ripeness at the time of harvest, processing, and storage [5]. Another important factor that determines the yield and type of substances that can be obtained is the extraction method. There are many different methods that can be used to extract phytochemicals from plant materials. Some of the conventional methods include heat reflux extraction, maceration, Soxhlet extraction, infusion, percolation, or decoction. [65,66]. Apart from choosing the extraction method, the type of eluent must be carefully selected as well. It should be chosen in accordance with the hydrophilic or amphiphilic nature of the target compound, as well as other criteria [67]. Generally, methanol is better for extraction of polyphenols with low molecular weight, while aqueous acetone is more efficient in extraction of flavanols that have higher molecular weight. Anthocyanidins, in turn, can be obtained with weak organic acids, like acetic acid or formic acid [66]. Different combinations might be used to get specific results. For instance, the yield of phenolics extracted from berry seed meal was highest when a combination of methanol, acetone, and water (7:7:6, *v*/*v*/*v*) was used, while acetone-water (80:20 *v*/*v*), methanol-water, and water gave smaller amounts [6]. Naturally, the use of different eluents will result in different chemical content of the extract, which in turn will cause differences in biological activity. For example, the ethyl acetate fraction of sea buckthorn seed extract has been shown to possess significantly higher antioxidant activity than the *n*-hexane fraction [8]. Other criteria that influence the results of phytochemical extraction include temperature, extraction time, and purification procedures. Increasing the temperature and extraction time might lead to higher yield, because of improved solubility, and decreased surface tension and viscosity. However, many phytochemicals can become hydrolyzed in higher temperatures, so extra care must be taken to choose the optimal conditions. Purification is often needed to get rid of unwanted materials, like proteins, carbohydrates, or lipids. Different purification strategies include solid phase extraction (SPE), column chromatography, or liquid-liquid extraction [66]. In recent years, many modern green extraction methods have been developed as well. Green extraction methods use less or no organic solvents to minimize impact on the environment and reduce costs. Examples include microwave and ultrasound-assisted extraction, supercritical fluid extraction, pressurized liquid extraction, enzyme-assisted extraction, and pulsed electric field extraction [65,66]. Newly obtained preparations are usually preserved by freeze-drying, so they can be stored for further use [67]. Extraction of new compounds should be followed by analysis of their structure and employment of appropriate molecular fingerprinting methods, to ensure that the compounds studied by different researchers are truly the same compounds, and enable in silico discovery of the relation between chemical structure and biological activity, which is an important part of drug development [68].

## 3. Bioavailability and Dosage

One of the factors that can impair in vivo activity of a compound is low bioavailability [69,70]. Unfortunately, this problem is apparent in many phytochemicals. Flavonoids usually have poor bioavailability, although it varies among different subclasses and individual compounds. Quercetin glycosides, isoflavones, and catechins are among the better-absorbed flavonoids, while anthocyanins are on the other side of the spectrum. For example, bioavailability of quercetin is 16–27.5%, luteolin 4.10–26%, kaempferol 2–20%, rutin 8.2%, apigenin 3–5%, and isorhamnetin 2.64% [71]. Bioavailability of phenolic acids has not been studied as often as that of flavonoids. Gallic and ferulic acids have good absorption rates, but there is still the issue of subsequent modifications of the absorbed compound’s chemical structure. For example, after absorption of ferulic acid, its free form in plasma constituted only 5–24%, while the rest was metabolized, mainly to sulfates and sulfglucuronides [72]. Bioavailability of phytosterols is quite poor. For example, only 1.9% of campesterol and 0.51% of sitosterol could be absorbed into the body [47].

Although poor bioavailability is an obstacle that can make compounds unsuitable for practical applications, it does not automatically disqualify it from in vivo use [73]. Oftentimes, compounds can be modified to increase their absorption into cells. For example, polymeric proanthocyanidins, which cannot be easily assimilated due to their high molecular size, can be converted into smaller molecules, for example by acid-mediated depolymerization in the presence of captopril [69]. Some phytochemicals can also be combined with each other—for example, resveratrol’s bioavailability can be enhanced by bonding it with piperine [74]. Moreover, various plant-derived substances can be combined with already existing drugs in order to improve their properties or decrease adverse effects [69]. Plant extracts and active substances can also be enclosed in lipid vesicles, polymeric nanoparticles, or other types of nanomaterials to improve their assimilation into the human body [75,76].

Even when bioavailability of a compound is high, its low content in food products might make assimilating the required dose through normal consumption impossible or not practical. Furthermore, when a compound is ingested in a form of drug or supplement, the dose required for achieving the desired effect might be toxic. For this reason, many compounds that showed good activity in vitro might not be viable for human applications. 

Converting the doses used in animal studies to human equivalent doses (HED) can be done with the help of various methods. Two such methods are “no observed adverse effect level” (NOAEL) and “minimum anticipated biological effect” (MABEL). Dose adjustment must include different metabolic rates between large and small animals, as well as pharmacokinetics. After the initial calculation, the dose is further modified to minimalize the risk of adverse effects. For the purpose of this review, we will convert animal doses to HED. We will use a method based on the differences in body surface area (as in [77]), though it must be noted that this is a simplified approach, and if these compounds or preparations would be used in clinical trials, the doses would have to be adjusted further. Moreover, this method is not suitable for drugs administered by subcutaneous, intramuscular, topical, or nasal routes, as well as proteins with molecular weight > 100 kDa [77,78].

## 4. Chemical Content of Seeds Eaten with Whole Fruit and Biological Activity of Seed Extracts, Fractions, Oil, Proteins, Lipids, Polysaccharides, and Isolated Compounds

### 4.1. Blackberry (Rubus fruticosus L.)

#### 4.1.1. Chemical Content of Blackberry Seeds

Blackberries (*Rubus fruticosus* L.) are shrubs belonging to the *Rosaceae* family, and are widely distributed in the northern hemisphere [79,80,81]. Blackberry seeds contain phenolic acids (ellagic acid, gallic acid, gallic hexoside), flavonoids (quercetin and its derivatives), anthocyanins (peonidin-3-glucoside), procyanidins (procyanidin B_1_), ellagitannins (sanguiin H-6), phytosterols (β-sitosterol), carotenoids, and tocopherols [6,11,29,79,82]. Ayoub et al. reported that blackberry seed meal had a higher concentration of total phenolics than some of the other similar plant species, like black raspberry and blueberry, which correlated with stronger antioxidant activity [6]. According to Choe et al., the major phenolics of the seed meal were ellagic acid (653.81 ± 66.84 μg/g) and sanguiin H-6 (457–675 μg ellagic acid equivalents/g) [11].

#### 4.1.2. Biological Activities of Extracts from Blackberry Seeds

##### Antioxidant Activity (In Vitro)

The antioxidant activity of various blackberry seed extracts was documented in multiple in vitro studies [6,11,29,79]. Wajs-Bonikowska et al. compared the radical-scavenging activity of several seed extracts depending on the type of eluent used in the extraction process. Ethanol extract was the most efficient at scavenging the ABTS^•+^ (2,2′-azino-bis-3-ethylbenzothiazoline-6-sulfonic acid) radical IC_50_ = 14.65 ± 0.90 μg/mL), while all the studied extracts—ethanol, carbon dioxide (CO_2_), and *n*-hexane—showed strong DPPH^•+^ (2,2-diphenyl-1-picrylhydrazyl) scavenging abilities (IC_50_ = 6.70 ± 0.40 μg/mL, 0.59 ± 0.01 μg/mL and 3.40 ± 0.14 μg/mL, respectively). It is worth noting, that the antioxidant activity of ethanol seed extract significantly surpassed the activity of extracts obtained from whole fruits of several blackberry cultivars [79]. Furthermore, non-polar fraction of the extract (100–1000 μg/mL) could significantly reduce intracellular reactive oxygen species (ROS) levels in IMR90, A549, Caco-2, and HepG2 cells after treatment with hydrogen peroxide (H_2_O_2_) [29]. The oxygen radical absorbing capacity (ORAC) assay indicated, that insoluble-bound phenolics had higher hydroxyl radical-scavenging abilities than free or esterified fractions. Insoluble-bound phenolics were also more efficient at preventing cupric ion-induced low-density lipoprotein (LDL) cholesterol oxidation. This was caused by the gradual release of phenolics from their insoluble-bound forms, which is illustrated by the increase in activity at longer incubation times. Blackberry seed meal extracts had higher ferric-reducing power than the extracts from black raspberry and blueberry meals. Here, the activities of free, esterified, and insoluble-bound phenolics were similar, which underlines the need for different assays in antioxidant studies [6]. 

##### Antimicrobial Activity (In Vitro)

Non-polar extract from blackberry seeds (obtained by Soxhlet extraction) had antibacterial activity against *Escherichia coli* (IAL 2064) and *Staphylococcus aureus* (ATCC 13565). The growth of *E. coli* was inhibited by 99.4%–33.4% (at concentrations of 3.3–0.42 μg/L), while the growth of *S. aureus* was inhibited by 90.7% and 33.3% (at concentrations of 33.3 and 1.67 μg/L, respectively). Interestingly, extracts obtained by Bligh-Dyer and ultrasound extraction methods did not show antimicrobial effects [29].

##### Antigenotoxic Activity (In Vitro)

Blackberry seed extract, as well as three compounds isolated from it—lambertianin C, sanguiin H-6, and 4-α-L-arabinofuranosylellagic acid—protected the DNA of human primary lymphocytes treated with an alkylating agent—mitomycin C. The extract from the ‘Thornfree’ cultivar at the concentration of 1 μg/mL had the strongest effect—it decreased the frequency of micronuclei formation by 62.4%. The effect was considerably stronger than that of amifostine (a cytoprotective and antioxidant agent), which reduced micronuclei formation by only 15.7%. The activity of most of the pure compounds isolated from the extract was comparable to that of amifostine. That suggests that the strong activity of the extract could be attributed to a synergistic effect of its components [82].

#### 4.1.3. Biological Activities of Blackberry Seed Flour, Polysaccharides, and Compounds

##### Modulation of Gut Microbiota by Blackberry Seed Flour (In Vitro)

In in vitro study conducted on bacteria extracted from a chow diet-fed C57BL/6J mouse feces, seed flour extract (0.4 g flour equivalent/mL) increased the number of *Bacteroidetes*, a phylum involved in the activation of T-cells, transformation of toxins, and bile acid metabolism. At the same time, the quantity of *Firmicutes* phylum was reduced. Some studies have found, that increased number of *Firmicutes* is connected to the process of aging [11]. The ratio between *Bacteroidetes* and *Firmicutes* might have an impact on other aspects of health, as researchers have found that obese mice had an increased number of *Firmicutes* and decreased number of *Bacteroidetes* [83]. Furthermore, seed meal extract increased the abundance of *Akkermansia* (thought to reduce the risk of obesity, diabetes, and inflammation), but reduced the number of *Bifidobacterium* and *Lactobacillus* (probiotic bacteria that are believed to have a positive impact on the health of their host). There is another aspect to the interaction between gut microbiota and blackberry seeds—the bacteria can metabolize ellagic acid to urolithins, compounds that have antioxidant, anti-inflammatory, and anticarcinogenic properties [11].

##### Modulation of Coagulation by Blackberry Seed Polysaccharides (In Vitro and In Vivo)

Blackberry seed polysaccharides affected coagulation—polysaccharide fractions BSP-1b, BSP-2, and BSP-3 had an antithrombotic effect in rabbit blood. BSP-1b could increase prothrombin time (PT), thrombin time (TT), and activated partial thromboplastin time (APTT), which indicates that its anticoagulant activity is tied to intrinsic and extrinsic pathways (as PT evaluates the efficiency of the extrinsic, and APTT intrinsic pathway). Extrinsic pathway is considered to be the first step in plasma-mediated hemostasis, and is activated by tissue factor present in the subendothelial tissue. Intrinsic pathway is a parallel pathway for thrombin activation, and begins with factor XII [80,84]. BSP-3 increased PT, TT, and APTT and decreased the amount of fibrinogen, while BSP-2 could prolong only TT and APTT, which indicates that it affects only the intrinsic pathway. One of the fractions—BSP-1a—had an opposite effect—it shortened PT, TT, and APTT and increased the amount of fibrinogen in the plasma. Anticoagulant properties of blackberry seed polysaccharides were further confirmed in vivo. 120 mg/kg of BSP-1b, BSP-2, or BSP-3 administered by gavage twice a day decreased the level of thromboxane B_2_ (TXB_2_) and increased the level of 6-keto-prostaglandin F1α (PGF_1α_) in rats. TXB_2_ and 6-keto-PGF_1α_ are used to reflect the levels of thromboxane A_2_ (TXA_2_) which is a platelet-activating agent, and epoprostenol (PGI_2_) which inhibits platelet function. The balance between TXA_2_ and PGI_2_ levels is an important factor in hemostasis. Another important anti-thrombotic compound is nitric oxide (NO). BSP-1b, BSP-2, and BSP-3 increased the concentration of endothelial nitric oxide synthetase (eNOS) and reduced the concentration of endothelin-1 (ET-1)—a peptide that can induce vascular dysfunction [80]. Human equivalent dose of 120 mg/kg in rats is approximately 19 mg/kg (HED mg/kg = 120 mg/kg × 0.162 = 19.44 mg/kg, method in [77]), which means that an average human (60 kg) would have to ingest 1140 mg of blackberry seed polysaccharides twice a day. The yield of BSP-1b, BSP-2, and BSP-3 extracted from 200 g of dried blackberry seeds was 21 mg, 50 mg, and 46 mg, respectively. Based on these preliminary calculations, it seems that it would be impossible to attain the amount required for biological effects only through dietary consumption, even when ignoring the additional problem of digestion and bioavailability. However, 1140 mg of extracted and isolated polysaccharides could be easily administered in the form of tablets or capsules [77,80].

##### Anti-Inflammatory Activity of Ellagic Acid and Ellagitannins (In Vitro and In Vivo)

Several studies have demonstrated anti-inflammatory properties of ellagic acid, a dimeric derivative of gallic acid abundant in blackberry seeds, and present in many different plant species [3,85,86,87].

In an in vitro study conducted on Jurkat T (human lymphocyte) cells stimulated with ionomycin or Phorbol-12-myristate 13-acetate (PMA), ellagic acid (30–60 μM) inhibited the expression of interleukin 2 (IL-2) and interferon γ (IFN-γ) [85]. In HaCaT (human, adult, low calcium, and high temperature) keratinocytes stimulated by tumor necrosis factor α (TNF-α) and interferon γ (IFN-γ), 250–1000 μM ellagic acid suppressed the expression of pro-inflammatory interleukin 6, and decreased relative mRNA expression of chemokines—thymic stromal lymphopoietin (TSLP), macrophage-derived chemokine (MDC), and thymus and activation-regulated chemokine (TARC). Furthermore, it inhibited pro-inflammatory Janus kinase/signal transducer and activator of transcription (JAK/STAT) and phosphoinositide 3-kinase/AKTsignaling pathways [86].

As for in vivo studies, 50 and 25 mg/kg of ellagic acid administered orally was reported to downregulate the expression of interleukin 6 (IL-6), interleukin 1β (IL-1β), TNF-α, and nuclear factor-kappa B (NF-κB) (an important transcription factor that regulates genes involved in inflammatory response) in the kidneys of rats treated with lead. In this case, human equivalent dose is approximately 8 and 4 mg/kg, which means, that an average person (60 kg) would have to ingest 480 or 240 mg of ellagic acid [3]. According to Choe et al., the content of ellagic acid in blackberry seed flour was 653.81 ± 66.84 μg/g, therefore 240 mg would be contained in over 360 g of seed flour, which is impractical for consumption. So, as was the case with blackberry seed polysaccharides, a sufficient dose would have to be provided in the form of a drug or extract with a higher concentration of ellagic acid [11]. In another study that assessed anti-inflammatory activity of ellagic acid, 40 mg/kg administered intraperitoneally could improve the condition of damaged skin, reduce the serum levels of immunoglobin (Ig)E, IL-6 and TNF-α, and decrease the infiltration of mast cells into skin lesions in murine atopic dermatitis model [86].

Other biological activities of ellagic acid include antioxidant, antiviral, anti-apoptotic, cardioprotective, cytotoxic, and anticarcinogenic [3,85]. Other compounds present in blackberry seeds also showed anti-inflammatory activity. For example, ellagitannins inhibited the transcription of NF-κB and decreased the secretion of interleukin 8 (IL-8) in an in vivo gastric inflammation model [88]. 

##### Activity of Sanguiin H-6 (In Vitro and In Vivo)

Sanguiin H-6 is an ellagitannin displaying a wide array of biological effects, found mainly in berries, including blackberry and black raspberry. Apart from its ability to protect DNA from mitomycin C-induced damage, Sanguiin H-6 was reported to have antioxidant, antibacterial, antitumor, anti-angiogenic, estrogenic, and anti-osteoclastogenic activity [2,48,49,50,51,52]. It could protect rat renal mitochondria from oxidative damage induced by peroxynitrite, decreasing the number of apoptotic cells, and reducing caspase-3 activity by 72% [89]. In murine primary osteoclasts, it decreased the concentration of ROS and upregulated the expression of heme oxygenase-1 (HO-1), a protein that protects cells from oxidative stress [49]. Furthermore, it inhibited the development of the biofilm of methicillin-resistant *Staphylococcus aureus* in a concentration-dependent manner [52]. As for the antitumor activity, 50–100 μM sanguiin H-6 increased the rate of apoptosis of human breast carcinoma cells (MDA-MB-231 and MCF-7) by 33.7–40.7%. In line with the results of Yokozawa et al., it could increase the activity of caspase-3 in both cell lines, as well as caspase-8 in MCF-7 and caspase-8 and -9 in MDA-MB-231 cells [50]. It inhibited the proliferation of A2780 (human ovarian carcinoma) cells as well. At the same time, it did not exert cytotoxic effect on non-tumorigenic LLC-PK1 porcine kidney cells [90]. On the other hand, high concentration of sanguiin H-6 (100 μM) had an opposite effect on cancer cells—it stimulated the proliferation of MCF-7 cells by binding to estrogen receptor α (Erα). This raises concern over its use and suggests that thorough testing is needed to ascertain if this effect occurs in vivo as well [51]. Additionally, Lee and Lee have found, that sanguiin H-6 has an antiproliferative effect on endothelial cells stimulated with vascular endothelial growth factor (VEGF) (IC_50_ ≈ 7.4 μg/mL) but does not affect the growth of HT1080 (human fibrosarcoma) cells [48]. Sanguiin H-6 could reduce the rate of bone resorption as well. 10 μg/body weight (g)/day injected intraperitoneally inhibited TNF-α-induced osteoclastogenesis by affecting RANKL (receptor activator of nuclear factor κB) cellular signaling and decreased the bone-resorbing activity of mature osteoclasts [49].

In a rat model of renal reperfusion injury, sanguiin H-6 (10 mg/kg/day, 30 days, administered orally) limited mitochondrial oxidative damage, decreased the number of apoptotic cells, and improved renal function. Human equivalent dose would be approximately 1.6 mg/kg (96 mg/day for a 60 kg person) [89]. Its antibacterial properties against methicillin-resistant *Staphylococcus aureus* were observed in vivo as well. 20 μL of 0.5 mg/mL topically administered sanguiin H-6 decreased wound infection in mice by 77.2–85.3% compared to control [52]. Inhibition of bone resorption was also studied on an animal model—the proportion of eroded surface to bone surface of mice injected with TNF-α was decreased in the group treated with sanguiin H-6 at the dose of 10 μg/body weight (g)/day, injected intraperitoneally [49].

### 4.2. Black Raspberry (Rubus coreanus Miq.)

#### 4.2.1. Chemical Content of Black Raspberry Seeds

Black raspberry (*Rubus coreanus* Miq.), known as Bokbunja in Korea, is a deciduous shrub distributed in South Asia, especially Korea, China, and Japan [91,92]. In Korea, the fruits are used to make wine [91]. Black raspberry seeds contain phenolic acids (gallic acid, ellagic acid, 3,4-dihydroxybenzoic acid), flavonoids (catechins) triterpenosides, ellagitannins (sanguiin H-6), and anthocyanins (cyanidin-3-rutinoside) [91,92,93]. According to Choi et al., total flavonoid content of the seeds was 1.31 g gallic acid equivalent per 100 g dry mass [93]. Omega-6 fatty acids (mainly linoleic and γ-linolenic acid) comprise more than 90% of total oil content [94].

#### 4.2.2. Biological Activities of Extracts from Black Raspberry Seeds

##### Antioxidant Activity (In Vitro)

Black raspberry seed extract had antioxidant (DPPH^•+^ radical- and hydrogen peroxide-scavenging) activity (IC_50_ = 4.58 μg/mL), however it was lower than that of caffeic acid, quercetin, and (+)-catechin (IC_50_ = 1.76 μg/mL, 1.53 μg/mL, and 2.17 μg/mL, respectively). The ability to scavenge superoxide anions was more potent—the activity of extract made from seeds discarded during wine production was lower than that of quercetin but higher than (+)-catechin. The efficiency of iron (Fe) (III) chelation was comparable to that of tannic acid [91]. The ORAC values of fresh seed extract and wine seed extract at the concentration of 50 μg/mL were 1041.9 and 1060.4 μM Trolox equivalents (TE)/g, respectively. The extracts protected against protein oxidation, lipid peroxidation, inhibited the formation of intracellular ROS by 20.9–29.7%, and protected against ROS-induced DNA damage [93]. RC_50_ values of ethanol seed extract in DPPH and ABTS assays were 26.68 go/mL and 39.30 μg/mL, respectively. The ferric reducing antioxidant power (FRAP) of the extract was 0.61 ± 0.01 mM ferrous sulfate (FeSO_4_)/mg [95]. 

##### Antiviral Activity (In Vitro and In Vivo)

Black raspberry seeds showed in vitro antiviral properties against influenza virus, norovirus-1, and feline calicivirus-F9 [92,96]. 50 μg/mL of seed extract inhibited plaque formation of BR59 (A/Brisbane/59/2007(H1N1)), KR01 (pandemic A/Korea/01/2009(H1N1)), and BR10 (A/Brisbane/10/2007(H3N2)) influenza virus strains by 93%-98% (in MDCK (Main-Darby canine kidney) cells). Pre-treatment with low molecular weight fraction (<1 kDa) of the extract caused complete, and co-treatment almost complete inhibition of plaque formation. Gallic acid also reduced the formation of virus plaque by 59–93% (at the concentrations of 1–400 μM), however other polyphenols isolated from the seeds had less significant effects. Furthermore, the low molecular weight fraction inhibited hemagglutination (at minimum inhibitory concentration of 0.01–0.1 μg/mL) and syncytium formation (at concentrations of 0.1–1 mg/mL). It showed in vivo activity as well—treatment with the doses of 1, 3.5, and 15 mg/kg/day administered orally for 5 days reduced the distribution of PR8 (A/Puerto Rico/8/1934(H1N1)) influenza virus in the lungs of infected mice and increased the survival rate to 100%. Human equivalent doses for a 60 kg person would be approximately 5, 17, and 73 mg [92]. In another study, seed extract and its low molecular weight fraction showed in vitro activity against murine norovirus-1 and feline calicivirus-F9, which are used as human norovirus surrogates. Gallic acid and cyanidin-3 glucoside also inhibited viral infection (by 50–65%). Low molecular weight fraction and cyanidin-3 glucoside bound to murine norovirus-1 polymerase, inhibiting the expression of viral genes, which is likely the cause of their antiviral activity [96].

#### 4.2.3. Hepatoprotective Activity of Black Raspberry Seed Oil (In Vitro)

Seed oil had hepatoprotective activity. It protected human HepG2 (hepatocyte carcinoma) cells from oxidative damage induced by H_2_O_2_. Incubation with the oil (50–200 μg/mL) reduced the levels of ROS and stimulated the expression of antioxidant enzymes—catalase (CAT), superoxide dismutase (SOD), and glutathione peroxidase (GPx). Increased expression of CAT, SOD, and GPx was the result of inhibition of extracellular signal-regulated kinase (ERK) and Jun *N*-terminal kinase (JNK)—enzymes that play a key role in regulating stress response and apoptosis [94].

### 4.3. Grape (Vitis vinifera L.)

#### 4.3.1. Chemical Content of Grape Seeds

Grape (*Vitis vinifera* L.) is a well-known plant with great economic importance, cultivated in temperate regions all around the globe. Grape seeds, which are a byproduct of wine and juice production processes, are a source of numerous bioactive substances [97,98,99]. 60% to 70% of total polyphenols contained in grape fruits are stored in the seeds [97]. Flavonoids present in the seeds include catechins, kaempferol, quercetin, rutin, and luteolin [1,5,97]. Grape seeds contain large amounts of proanthocyanidins (including procyanidins, e.g., procyanidin B_2_), as well as anthocyanins (cyanidin-3-glucoside), phenolic acids (chlorogenic acid, caftaric acid, caffeic acid), and stilbenes (reservatrol) [5,53,100,101]. The fermentation processes that occur during wine production alter the content of phytochemicals. For example, the concentration of polymeric products, like proanthocyanidins or oligostilbenes increases significantly [97].

#### 4.3.2. Biological Activity of Extracts from Grape Seeds

##### Antioxidant Activity (In Vitro and In Vivo)

Grape seeds have robust antioxidant properties [5,99,100,102]. In a study by Tang et al., the FRAP values ranged from 312.4 ± 11.8 to 837.2 ± 21.6 μmol Fe(II)/g fresh weight (FW), depending on the cultivar and region. The Trolox equivalent antioxidant capacity (TEAC) values varied between 207.8 ± 10.6 and 473.5 ± 19.3 μmol Trolox/g FW. The FRAP and TEAC values were higher for seeds than for skin [5]. Bosso et al. studied the effect of maceration time on antioxidant activity of the seeds from four different grape cultivars used in wine production. After short maceration (2 days) the ‘Uvalino’ variety had the highest ABTS, DPPH, and FRAP values, followed by ‘Grignolino’, ‘Nebbiolo’, and ‘Barbera’ varieties. After racking off, the highest values were recorded in the ‘Grignolino’ variety. Maceration decreased the antioxidant activity of all grape varieties. For example, the initial ABTS value for ‘Uvalino’ was 52.5 mmol TE/100 g DW, and after racking off it was 20.3 mmol TE/100 g dry weight (DW) [103].

In a clinical trial on 32 subjects with type II diabetes mellitus and elevated risk of cardiovascular disease, Kar et al. measured the effect of grape seed extract (300 mg twice a day administered for 4 weeks) on oxidative stress markers. Reduced glutathione concentrations increased (from 2630 ± 823 μM to 3595 ± 1051 μM), however glutathione ratio and plasma total antioxidant status (TAOS) remained unchanged [104]. A meta-analysis of 19 clinical trials examining the effect of grape seed extract (GSE) on oxidative stress showed that it significantly decreased the level of malondialdehyde (MDA) and oxidized LDL, while total antioxidant capacity was only slightly increased [105].

##### Anti-Inflammatory Activity (In Vivo)

In a clinical trial by Kar et al. mentioned in the previous section, 300 mg of grape seed extract twice a day (taken for 4 weeks) decreased the levels of highly sensitive C-reactive protein (hsCRP)—an inflammatory marker useful for cardiovascular events prediction—in subjects with diabetes and heightened risk of cardiovascular disease [104,106]. A meta-analysis by Foshati et al. that included 7 clinical trials (total number of participants—277) also showed that GSE supplementation could decrease hsCRP concentration (−0.48 mg/L, 95% confidence interval (CI): −0.94, −0.03) [107].

##### Inhibition of Amyloid β Oligomerization (In Vitro)

A commercially available grape seed phenolic extract (MegaNatural-AZ^®^ (Polyphenolics, Madera, CA, USA)) could inhibit the oligomerization of amyloid β (Aβ). Aβ oligomerization is a key process in the development of Alzheimer’s disease. Monomeric, dimeric, and oligomeric fractions of the extract were studied as well—monomeric fraction had the most potent inhibitory activity when mass concentration was considered, while oligomeric fraction was the most potent while the molar concentration was considered [12].

##### Anti-Proliferative and Pro-Apoptosis Activity Cancer Cells (In Vitro)

Grape seed extract inhibited the growth of OVCAR-3 (human ovarian cancer) cells (IC_50_ = 71 μg/mL). The number of apoptotic cells increased, as did the expression of caspase 3. Incubation with the extract caused upregulation of tumor suppressor genes *PTEN* and *DACT1.* The results indicated that treatment with the extract increases the expression of pro-apoptotic *BAX* and decreases the expression of anti-apoptotic *BCL2* [108].

##### Cardioprotective Activity (In Vitro and In Vivo)

Grape seeds contain substances that have various beneficial effects on the cardiovascular system [102,109,110,111,112]. In a randomized, double-blind, placebo-controlled study by Schön et al., grape seed extract (GSE) (Enovita^®^ (Indena, Tours, France)) at the dose of 150 mg/day decreased blood pressure in mildly hypertensive subjects. 16 weeks administration resulted in reduced systolic (−4.6 mmHg, 95% CI: −6.9 to −2.3) and diastolic (−3.2 mmHg, 95% CI: −5.1 to −1.4) blood pressure, which was measured by the patients over the span of 7 days. Interestingly, the decrease was observed only in men, while in women there were no significant differences between GSE and placebo. However, a 24 h measurement showed significant effects in both men and women. Moreover, the extract increased nocturnal dipping, which is a physiological decrease in blood pressure during the night, positively correlated with decreased risk of cardiovascular mortality [113]. In a meta-analysis of 9 clinical trials assessing the effects of grape seed extract, Feringa et al. have reported a statistically significant decrease in systolic blood pressure (−1.54 mmHg) and heart rate (−1.42). Even though the reduction of these parameters was comparatively small, the authors have proposed, that grape seed extract might still be useful for lowering the risk of all-cause mortality, mortality after stroke, and mortality after coronary artery disease, as a reduction of systolic blood pressure by only 3 mmHg has been estimated to diminish the risk of these events by 4%, 8%, and 5%, respectively. On the other hand, the analysis did not show any significant reductions in diastolic blood pressure or lipid levels [107].

##### Hypoglycemic Activity (In Vitro and In Vivo)

Grape seed extract turned out to be a potent α-glucosidase inhibitor, with IC_50_ and IC_90_ values lower than acarbose, although more research is needed to determine if its inhibitory activity is equally effective in vivo. Since currently used inhibitors (e.g., acarbose, miglitol) have adverse side effects, finding new substances that can have the same therapeutical effect would be beneficial [114]. 

In a double-blind randomized placebo-controlled trial, Kar et al. measured the effect of grape seed extract on glycemia biomarkers in a group of 32 male and female subjects with type II diabetes mellitus and high risk of cardiovascular disease. Administration of 300 mg twice a day over a period of 4 weeks resulted in lower fructosamine levels, but the changes in fasting glucose level and insulin resistance were not significant. Additionally, the researchers have noted that total cholesterol concentration decreased, while high-density lipoprotein (HDL) cholesterol and triglyceride levels remained unchanged [104]. In a study that included 42 adolescents, supplementation of GSE (100 mg/day) for 8 weeks significantly reduced insulin levels and insulin resistance (measured by Homeostatic Model Assessment for Insulin Resistance (HOMA-IR)) [115]. Asbaghi et al. analyzed the results from 15 clinical trials that studied the effect of grape seed extract on glycemic control. At the dose of 300 mg a day or higher, GSE reduced the levels of fasting plasma glucose. On the other hand, it did not have any effect on the concentration of glycated hemoglobin A1c (HbA1c) [116].

##### Antibacterial Activity (In Vitro)

Two extracts from the seeds of ‘Bangalore’ variety had antibacterial activity against *B. cereus*, *B. subtilis*, *S. aureus*, *B coagulans*, *E. coli*, and *P. aeruginosa.* The minimal inhibitory concentration values (MIC) of the methanol:water:acetic extract were 900 ppm for *B. cereus*, *B. subtilis*, and *B. coagulans*, while the values for *S. aureus*, *E.coli*, and *P. aeruginosa* were 1000 ppm, 1250 ppm and 1500 ppm, respectively. MIC of the acetone:water:acetic extract for *B. cereus*, *B. subtilis*, and *B. coagulans* was 850 ppm, while in the rest of bacteria the inhibitory activity was the same in both extracts. The extracts were more effective against Gram-positive bacteria [117]. This is consistent with the results of other authors, who have noticed that extracts from grape pomace show weaker inhibition against Gram-negative bacteria. This phenomenon is attributed to the presence of polysaccharide cell wall, which can limit the penetration of polyphenols into the cell [118]. Grape seeds inhibited the growth of *Helicobacter pylori* as well. The MIC values of muscadine seed extract varied from 256 to 1024 μg/mL in different strains, though the skin extract was more effective in some of them (MIC = 256–512 μg/mL) [119]. 

##### Modulation of Gut Microbiota (In Vivo)

Commercial GSE facilitated the recovery of gut microbiota caused by antibiotic treatment in mice fed with high-fat diet. 200 mg/kg of GSE increased the abundance of *Verrucomicrobia* and decreased the amount of *Actinobacteria*. Moreover, it partially restored the content of *Akkermansia* in the feces and increased the relative abundance of *Alloprevotella* from 0.0075% to 0.0113%. Human equivalent dose of GSE would be approximately 16 mg/kg, which means that a 60 kg person would have to ingest 960 mg [120].

#### 4.3.3. Biological Activity of Grape Seed Oil and Compounds

##### Antioxidant Activity of Grape Seed Proanthocyanidins, Oligomeric Procyanidins, and Oil (In Vitro)

He et al. showed, that seed proanthocyanidins (5–25 μM) could decrease the concentration of reactive oxygen species in PC12 (rat adrenal pheochromocytoma 12) cells, mitigating the damage induced by hydrogen peroxide, and significantly increasing their viability [100]. Oligomeric grape seed procyanidins upregulated the NF-E2-related transcription factor (Nrf2) pathway and HO-1 expression in HEK293 (human embryonic kidney) cells. Nrf2 plays an important role in regulating antioxidant response by activation of various antioxidant enzymes, including HO-1. Upregulation of HO-1 exerted a protective effect on HEK-293 cells damaged by cisplatin [121,122]. The unsaponifiable fraction of grape seed oil (10–100 μg/mL) could suppress lipopolysaccharide (LPS)-induced production of ROS and NO in human primary monocytes. The expression of nitric oxide synthase (Nos2) was decreased. In line with the results from Tang et al., seeds had stronger antioxidant activity than the skin, as well as other grape parts and products, like leaves and wine [5,123]. 

##### Anti-Inflammatory Activity of Unsaponifiable Fraction from Grape Seed Oil (In Vitro)

The unsaponifiable fraction of grape seed oil (consisting mostly of β-sitosterol, tocopherols, and tocotrienols) showed anti-inflammatory activity in human primary monocytes. It downregulated the expression of pro-inflammatory genes (Toll-like receptor 4 (*TLR-4*), *TNF-alpha*, *IL-1 beta*, and *IL-6*), which resulted in reduced production of TNF-α, IL-1β, and IL-6. At the same time, it decreased the concentration of arachidonic acid, which is responsible for production of prostaglandins and leukotrienes. Its ability to regulate the balance between CD14 and CD16 monocytes was yet another way of suppressing the inflammatory response, as a disproportion between different monocyte subsets can promote persistent inflammation [123].

##### Anti-Apoptotic Activity of Grape Seed Proanthocyanidins and Procyanidins (In Vitro and In Vivo)

Grape seed proanthocyanidins (5–10 μM) had an anti-apoptotic effect on PC12 cells damaged by hydrogen peroxide. The viability significantly improved and the ratio of cells that underwent apoptosis decreased. The results suggested that this effect could be due to upregulation of the phosphatidylinositol 3-kinase (PI3K)/AKT signaling pathway [100]. In a diabetic neuropathy model, procyanidin B_2_ (10 μg/mL) suppressed the apoptosis of rat mesangial cells (HBZY-1) treated with a high dose of glucosamine. This could be attributed to the amelioration of ROS-induced cellular damage and mitochondrial dysfunction, as well as prevention of glucosamine-induced decrease in the expression levels of phosphor-AMP-activated protein kinase (AMPK), sirtuin 1 (SIRT1), and peroxisome proliferator-activated receptor gamma coactivator 1-alpha (PGC-1α) by procyanidin B_2_ [53]. Procyanidin B_2_ can also inhibit dipeptidyl peptidase 4 (DPP4). Studies have found that the level of DPP4 is elevated in several diseases, and its inhibition decreases the inflammatory response and oxidative stress. Procyanidins increased cell viability and decreased the concentration of pro-apoptotic proteins (cleaved-caspase 3 and Bax) in rat primary chondrocytes stimulated by IL-1β. At the same time, the concentration of Bcl2, an anti-apoptotic protein, increased. The number of apoptotic and senescent chondrocytes was reduced. The study has also revealed that this apoptosis- and senescence-suppressing activity is related to the inhibition of DPP4 and upregulation of SIRT1, which is an aging- and senescence-related protein. Furthermore, DPP4 inhibition diminished progression of osteoarthritis in an in vivo mouse model. 40 mg/kg of grape seed procyanidins administered orally twice a week over the course of 8 weeks significantly decreased the number of senescent and apoptotic cells, alleviating the changes in articular cartilage. Human equivalent dose (approximately 3.2 mg/kg) for an average, 60 kg person would be 192 mg [124].

##### Anti-Proliferative and Pro-Apoptosis Activity of Grape Seed Proanthocyanidins toward Cancer Cells (In Vitro)

Grape seed proanthocyanidins inhibited the proliferation and induced apoptosis of PANC-1 (human pancreatic cancer) cells. This effect was attributed to modulation of miRNA expression levels [125]. Furthermore, grape seed extract inhibited the growth of OVCAR-3 (human ovarian cancer) cells (IC_50_ = 71 μg/mL). The number of apoptotic cells increased, as did the expression of caspase 3. Incubation with the extract caused upregulation of tumor suppressor genes *PTEN* and *DACT1.* The results indicated that treatment with the extract increases the expression of pro-apoptotic *BAX* and decreases the expression of anti-apoptotic *BCL2* [108].

##### Cardioprotective Activity of Grape Seed Procyanidins, Proanthocyanidins, and Polyphenols (In Vitro and In Vivo)

In an in vitro study by Shao et al., seed procyanidins prevented macrophage foam cells formation, which is a process linked to the pathophysiology of atherosclerosis. Procyanidins decreased the expression of macrophage lipid absorption receptors (scavenger receptor B (CD36), scavenger receptor A (SR-A), and lectin-like ox-LDL receptor 1 (LOX-1)), while the expression of cholesterol efflux-related receptor—phospholipid-transporting ATPase (ABCA1)—increased. Their effectiveness varied depending on the concentration and incubation time. 25 μg/mL of procyanidins incubated with the cells for 48 h gave best results. Moreover, they downregulated acetyl coenzyme A (CoA) acetyltransferase (ACAT1), an enzyme that facilitates accumulation of cholesterol in foam cells [102]. Polyphenolic grape seed extract had antiplatelet and anticoagulant activities, while several major polyphenols present in grape seeds—catechin, epicatechin, and gallic acid—had a hypocholesterolemic effect. They inhibited pancreatic cholesterol esterase (IC_50_ > 100 μg/mL) which impairs the absorption of dietary cholesterol and bonded to bile acids (taurocholic acid and glycodeoxycholic acid), which is thought to be a molecular mechanism that helps decrease blood cholesterol levels [109]. 

The cardioprotective activity of grape seeds was studied in vivo as well. Grape seed proanthocyanidin extract (250 mg/kg/day) significantly decreased rat systolic blood pressure and partially prevented vascular remodeling caused by long-term use of ouabain, an anti-hypotensive drug. Human equivalent dose is 40.5 mg/kg, which means, that an average, 60 kg person would have to ingest approximately 2400 mg of the extract [111]. Safety assessment conducted on healthy volunteers over the period of four weeks showed no major adverse effects up to a dose of 2500 mg/day [126]. 

In a clinical trial by Odai et al. pre-hypertensive, 40–64 years old Japanese men and women were administered tablets with grape seed proanthocyanidins at the dose of 400 mg/day. After 12 weeks of intervention, mean systolic and diastolic blood pressure was reduced by 13.1 and 6.5 mmHg, respectively. Furthermore, the researchers noted improvement in vascular elasticity in non-smoking participants [127].

##### Modulation of Lipid Metabolism by Grape Seed Procyanidins (In Vitro)

Grape seed procyanidin extract changed the expression of adipogenic- and lipolytic-related genes, which led to stimulation of lipolysis, and reduced accumulation of lipids and triglycerides during the differentiation of porcine primary adipocytes. The differentiation of preadipocytes was suppressed, and the cell cycle was arrested. High concentrations (200 and 300 μg/mL) increased the rate of apoptosis as well [128].

##### Hypoglycemic Activity of Grape Seed Oil (In Vitro)

Phenolic fraction of grape seed oil inhibited protein tyrosine phosphatase 1B (PTP-1B), an enzyme that downregulates the insulin and leptin receptor signaling pathways and is overexpressed in type II diabetes. Oil pressed from the seeds of ‘Sauvignon Blanc’, ‘Riesling’, ‘Syrah’, ‘Chenin Blanc’, and ‘Cabernet Sauvignon org’ varieties exhibited the highest inhibitory activity. As inhibition of PTP-1B increases insulin sensitivity and helps control glycemia, there is a great interest in finding new substances that have this effect [98]. 

##### Modulation of Gut Microbiota by Grape Seed Proanthocyanidins (In Vivo)

Seed proanthocyanidins influenced the gut microbiota of female rats fed with a standard diet. A dose of 500 mg/kg decreased the number of Firmicutes and increased the number of Bacteroidetes and Proteobacteria. The abundance of *Bacteroidaceae* and *Porphyromonadaceae* was increased considerably. In contrast, the number of *Ruminococcacea* and *Dehalobacteriaceae* was decreased in comparison to control. In this case, HED is approximately 80 mg/kg (4800 mg for a 60 kg person) [129].

### 4.4. Moringa oleifera Lam.

#### 4.4.1. Chemical Content of *M. oleifera* Seeds

*Moringa oleifera* is a perennial tree native to India, widely distributed in numerous tropical and subtropical countries [7,59,130]. Seed kernels contain up to 40% of oil, which consists of >70% oleic acid. *M. oleifera* seeds contain isothiocyanates (mainly 4-[(α-L-Rhamnosyloxy) benzyl] isothiocyanate—moringa isothiocyanate-1, also called moringin), flavonoids (catechin, epicatechin, quercetin, kaempferol), phenolic acids (mostly gallic, ellagic, and caffeic acids), phenolic glycosides (niazirin), phytosterols (β-sitosterol, stigmasterol, campesterol), alkaloids, glucosinolates, thiocarbamates, and large amount of tocopherols (α-, γ- and δ-tocopherols) [7,31,39].

#### 4.4.2. Biological Activities of *M. oleifera* Seed Extracts

##### Antioxidant Activity (In Vivo)

Seed extract at the dose of 50–200 mg/kg/day activated the Nrf2/HO-1 signaling pathway in diabetic rats, protecting their kidney function [131].

##### Anti-Inflammatory Activity (In Vivo)

Moringa seed extract enriched with MIC-1 (500 mg/kg) reduced carrageenan-induced rat paw edema by 33%, which was comparable to aspirin at the concentration of 300 mg/kg (27%) [31]. 

##### Neuroprotective Activity (In Vivo)

*M. oleifera* has neuroprotective activity—seed extract (500 mg/kg) protected mice from cerebral injury caused by ischemic stroke. The best results were obtained when it was administered two hours after reperfusion. The study has revealed, that this effect was linked to upregulation of expression of brain-derived neurotrophic factor (BDNF), neurotrophin-3 (NT3), and beta-nerve growth factor (NGF), which are proteins related to neurogenesis [132]. Moreover, the extract at the concentration of 500 mg/kg could protect mice from the impairment of mental functions induced by scopolamine. It ameliorated the degeneration of memory and learning processes, and prevented the decrease in acetylcholine levels by inhibition of acetylcholinesterase. In this case, human equivalent dose would be approximately 40 mg/kg [133].

##### Cytotoxic Activity toward Cancer Cells (In Vitro)

In a study by Aldakheel et al., seed extract (30–100 μg/mL) significantly reduced the viability of HCT-116 (human colon cancer) cells, while the survival rates of non-tumorigenic HEK-293 (human embryonic kidney) cells remained unaffected [134]. Adebayo et al. assessed the effect of various seed extracts and fractions on MCF-7 (breast cancer) cells. Hexane and dichloromethane fractions of crude ethanolic extract inhibited cell growth (IC_50_ = 130 μg/mL and 26 μg/mL, respectively) in a dose dependent manner. Crude water extract inhibited proliferation as well, but its efficiency was lower *(*IC_50_ = 280 μg/mL) [135].

##### Antibacterial Activity (In Vitro)

Seed extract showed antibacterial activity against *S. aureus* (at the concentrations of 50–250 μg/mL) and, to a lesser degree, *E. coli* (at the concentrations of 100–250 μg/mL) [134,136].

#### 4.4.3. Biological Activities of *M. oleifera* Seed Oil, Meal, Proteins, and Compounds

##### Antioxidant Activity of *M. oleifera* Seed Proteins (In Vitro)

Seed proteins had radical-scavenging activity against DPPH^•+^, ABTS^•+^, OH^•^, and O_2_^•−^ radicals, and could reduce lipid peroxidation. Enzymatic hydrolysis of the proteins with flavourzyme significantly increased their antioxidant activity—scavenging rate of DPPH^•+^ increased from 63.25% to 85%. Subsequent investigation of the correlation between the molecular size of the peptide and its antioxidant activity showed, that the fragments that weighted less than 3.5 kDa had stronger activity than peptides with a molecular weight of 3.5–5 kDa and larger [59]. *M. oleifera* seeds contain seven kinds of hydrophobic amino acids, namely alanine, valine, methionine, isoleucine, leucine, tyrosine, and phenylalanine. As hydrophobic amino acid content is an important factor that contributes to the antioxidant activity of bioactive proteins, it has been proposed that they might be one of the reasons for the radical-scavenging activity of *M. oleifera* seed proteins [59,62]. In another study, >1 kDa seed globulins could scavenge 64.24% of DPPH^•+^ radicals which was more than any of the other fractions with higher molecular mass. Seed proteins showed robust ferric-reducing and metal-chelating activity as well, but their hydroxyl radical scavenging capacity was comparatively lower [137]. 

##### Anti-Inflammatory Activity of *M. oleifera* Seed Oil (In Vivo)

Anti-inflammatory properties of *M. oleifera* oil were studied in vivo by Cretella et al. [138] and Jaja-Chimedza et al. [31]. Topically administered seed oil decreased ear inflammation and edema induced in rats by tetradecanoylphorbol-13-acetate (TPA) (by 69.2 ± 8.8%) or phenol (by 61.2 ± 10.9%), though the anti-inflammatory effect was due to interaction with glucocorticoid receptors, which raises concerns about the potential side effects [138].

##### Neuroprotective Activity of *M. oleifera* Seed Oil (In Vivo)

*M. oleifera* seed oil prevented methotrexate-induced cerebral neurotoxicity. The dose of 5 mL/kg body weight reduced the levels of oxidative stress and inflammation in the cerebrum of rats. Consistently with the previous study, the activity of acetylcholinesterase was decreased. The vacuolar changes and necrosis revealed by histopathological observations were less severe as well [139]. 

##### Cytotoxic Activity of *M. oleifera* Seed Oil and Lectin toward Cancer Cells (In Vitro and In Vivo)

Nano-micelle of *Moringa oleifera* seed oil significantly reduced the viability of MCF-7 *(*IC_50_ = 86.5 μg/mL), HCT 116 *(*IC_50_ = 49.1 μg/mL), and Caco-2 (concentrations of 60–100 μg/mL induced cell death in approximately 50% cells) cell lines [140].

Anticancer activity of *M. oleifera* seeds was studied in vivo as well. Seed lectin inhibited the growth of Ehrlich ascites carcinoma (EAC) cells. The mice were injected with EAC cell intraperitoneally, and then treated with seed lectin injections once a day for five consecutive days. Cell growth was inhibited by 15.27% (at the dose of 2 mg/kg) and 55% (at the dose of 4 mg/kg) [141].

##### Antibacterial Activity of *M. oleifera* Seed Meal Extract, and Lectin (In Vitro)

Seed meal extract obtained with 200 W ultrasonification power inhibited the growth of *S. aureus* and *E. coli* (MIC = 3.13 and 6.25 mg/mL). It was also effective against *S. typhimurium* (MIC = 25 mg/mL) and *B. cereus* (MIC = 3.13 mg/mL) [142]. Seed lectin, in turn, could prevent the development of *Bacillus sp.* and *S. marcescens* biofilm. Even low concentrations (0.325, 0.65, and 1.3 μg/mL) were effective against *S. marcescens.* Lectin reduced the growth of *Bacillus* sp. at all tested concentrations (0.65–41.6 μg/mL), but 20.8 and 41.6 μg/mL had especially strong activity, comparable to the antibiotic rifampicin [143].

##### Activity of MIC-1 (Moringin) (In Vitro and In Vivo) 

MIC-1 (moringin) is an isothiocyanate present in *M. oleifera* seeds. It has attracted attention due to its wide range of biological activities, including anti-inflammatory and pain-relieving. Researchers have also studied its therapeutic potential in neurodegenerative disorders [130,144,145]. In a study by Sailaja et al., the levels of ROS in the cytoplasm of LPS-induced RAW264.7 murine macrophages treated by 10 μM moringa isothiocyanate-1 (MIC-1) were decreased by 76%. The researchers have also discovered that MIC-1 triggers translocation of Nrf2 to the nucleus, which may be the reason for its antioxidant activity [146].

MIC-1 had antinociceptive properties as well [144,147]. Nociceptive pain results from acute noxious stimuli (e.g., chemical stimulation, heat, or mechanical force). When noxious stimulus persists long enough, nociceptive pain can transform into inflammatory pain through the release of pro-inflammatory factors from nociceptive neurons [148]. MIC-1 activated and desensitized transient receptor potential type ankyrin ion channel (TRPA1) in transfected HEK-293 (human embryonic kidney) cells. TRPA1 is involved in transduction of signal in response to different stimuli, including cold, pungent compounds, airborne irritants, and cannabinoids. Its agonists are known to decrease the sensation of pain [144]. These properties were also studied in vivo. 2% moringin cream administered topically alleviated neuropathic pain in mice with multiple sclerosis. The cream also had an anti-inflammatory effect [149]. Furthermore, in an in vitro study by Jaja-Chimedza et al., MIC-1 inhibited the production of NO in LPS-induced RAW 264.7 murine macrophages (97.5% at the concentration of 10 μM) and downregulated the expression of pro-inflammatory proteins (inducible nitric oxide synthase (iNOS), IL-1β and IL-6) [31]. The anti-inflammatory activity of MIC-1 was further confirmed in an in vivo study by Sailaja et al. Oral administration of 80 mg/kg MIC-1 in LPS-induced sepsis/acute inflammation mice model significantly reduced the expression of TNF-α, IFN-α, IL-1β, and IL-6 in the kidney, liver, colon, and spleen. These effects are possibly related to modulation of Nrf-2- and NF-κB-regulated genes—MIC-1 promoted the translocation of Nrf-2 to the nucleus and reduced the binding of NF-κB to *TNF-α* and *IL-6* promoters. Human equivalent dose would be approximately 6.5 mg/kg (390 mg for a 60 kg person) [146].

Cirmi et al. discovered that moringin (4-(α-L-rhamnopyranosyloxy) benzyl C) inhibited the growth of SH-SY5Y (human neuroblastoma) cells through induction of apoptosis. The greatest inhibitory effect of 73% was achieved by 72 h incubation at the concentration of 16.4 μM. At the same time, the viability of a non-tumorigenic WI-38 (human diploid fibroblast) cell line was not significantly reduced [130].

##### Activity of Niazirin (In Vitro and In Vivo)

Niazirin, a phenolic glycoside isolated from the seeds of *M. oleifera* inhibited in vitro proliferation of porcine primary vascular smooth muscle cells (VMSCs) induced by high glucose levels. During diabetes, increased level of oxidative stress caused by hyperglycemia can lead to abnormal proliferation of VMSCs, and in consequence, atherosclerosis. Niazirin reduced VMSCs growth, but only in high-glucose environment—it did not affect cells cultured in normal conditions. The effect was observed in vivo as well—it decreased the levels of ROS in the serum and inhibited overexpression of Ki67 (an indicator of proliferation) in aortas of diabetic mice [39]. In another in vivo study, niazirin (20 mg/kg/day, administered by gavage) decreased the food intake and body weight of db/db diabetic mice, improved hyperglycemia, ameliorated insulin resistance, and improved hepatic carbohydrate metabolism. Approximate human equivalent dose in this case is 1.6 mg/kg, 96 mg for a 60 kg person [150]. 

In the continuation of the above-mentioned in vivo study by Bao et al., anti-inflammatory properties of niazirin were revealed as well. It significantly decreased the concentration of plasma TNF-α and increased the levels of interleukin 10 (IL-10). IL-10 is an anti-inflammatory cytokine that reduces the release of inflammatory mediators, and subsequently inhibits migration and proliferation of inflammatory cells [150]. Moreover, it showed a strong inhibitory effect on the expression of several proinflammatory cytokines (IL-8, interleukin 12 (IL-12), interleukin 17 (IL-17), interleukin 22 (IL-22), and interleukin 23 (IL-23)) in vitro, in LPS-stimulated human THP-1 leukemia monocytes [151].

Apart from its hypoglycemic and anti-inflammatory effects, niazirin could moderate plasma lipid profiles. In an in vivo study on db/db mice, the dose of 20 mg/kg/day decreased the levels of LDL cholesterol (LDL-C), triglycerides, and non-esterified fatty acids, at the same time increasing the concentration of HDL cholesterol (HDL-C) [150].

### 4.5. Sea Buckthorn (Hippophae rhamnoides L.)

#### 4.5.1. Chemical Content of Sea Buckthorn Seeds

Sea buckthorn (*Hippophae rhamnoides* L.) (also classified as *Elaeagnus rhamnoides* (L.) A. Nelson) is a thorny, deciduous tree or shrub, distributed mainly in Europe and Central Asia [8,152,153]. Sea buckthorn seeds are encapsulated in bright orange berries, and contain flavonoids (isorhamnetin, kaempferol, quercetin), tocopherols, procyanidins, and phytosterols (mostly β-sitosterol) [8,154,155,156,157].

#### 4.5.2. Biological Activities of Sea Buckthorn Seed Extracts

##### Anti-Inflammatory and Hypoglycemic (In Vivo)

In mice fed with high-fat diet, flavonoid-enriched seed extract (100–300 mg/kg) administered orally partially alleviated glucose intolerance and decreased fasting blood glucose levels by 14.55%. Moreover, researchers have found, that it could reduce the infiltration of macrophages into the adipose tissue. Approximate human equivalent dose of 8–24 mg/kg would mean, that an average 60 kg person would have to ingest at least 480 mg of the extract [158]. 

##### Anti-Hyperlipidemic Activity (In Vivo)

Flavonoid-enriched seed extract reduced hypertriglyceridemia, hepatic triglyceride accumulation, and obesity in mice fed with high-fat diet. Body weight gains of mice supplemented with the extract were lower by 33.06% (at the dose of 100 mg/kg) and 43.51% (at the dose of 300 mg/kg). The adipocytes were smaller in size and the accumulation of triglycerides in the liver was 49.56% lower. The level of serum triglycerides decreased as well, but total cholesterol and HDL-C remained unaffected. These effects could be accredited to inhibition of peroxisome proliferator-activated receptor γ (PPARγ) expression in the liver. PPARγ is responsible for regulating the accumulation of lipids in white adipose tissue. Human equivalent dose would be approximately 8 and 24 mg/kg (480 and 1440 mg for a 60 kg person) [158].

##### Antibacterial Activity (In Vitro)

A study by Chauhan et al. revealed, that sea buckthorn seeds have antibacterial activity. Aqueous seed extract inhibited the growth of *Listeria monocytogenes* (by 100% at the concentration of 750 ppm) and *Yersinia enterocolitica* (by 100% at the concentration of 1000 ppm) [159]. Chloroform, acetone, and methanol seed extracts also showed antibacterial activity against *L. monocytogenes* and *Y. enterocolitica*, as well as *B. cereus*, *B. coagulans*, and *B. subtilis*. Methanol extract had the strongest effect, and *Y. enterocolitica* was the most resistant to the inhibitory activity of all three extracts. This resistance could be attributed to the presence of lipopolysaccharides in the membranes of Gram-negative bacteria [160]. Arora et al. studied the effect of the extracts from sea buckthorn leaves, seeds, and pomace on 17 different strains of Gram-positive and Gram-negative bacteria. Though the leaves showed the strongest antibacterial activity overall, the seeds were more effective than the pomace and strongly inhibited the growth of *Bacillus cereus* (inhibition zone = 17.7 mm), *P. aeruginosa* (15.3 mm), and *Salmonella enterica* (14.0 mm) [161].

#### 4.5.3. Biological Activity of Sea Buckthorn Seed Oil, Proteins, and Compounds

##### Antioxidant Activity of Sea Buckthorn Seed Oil, and Flavonoids (In Vitro and In Vivo)

Among the three main flavonoids from the seeds—isorhamnetin, kaempferol, and quercetin—quercetin had the strongest DPPH^•+^, ABTS^•+^, hydroxyl, and superoxide radical-scavenging ability, while their ferric-reducing capabilities were comparable [8]. Ethyl acetate fraction of the seed extract had a stronger effect than *n*-hexane extract [8]. In a clinical trial by Vashishtha et al., supplementation of healthy volunteers with seed oil capsules (0.75 mL/day) caused an increase in serum total antioxidant status (determined with the ABTS method) by 46.13 μmol Trolox equivalent/L [162].

##### Anti-Inflammatory and Hypoglycemic Activity of Sea Buckthorn Seed Flavonoids and Proteins (In Vitro and In Vivo)

Two in vitro studies have shown hypoglycemic properties of sea buckthorn seeds. Fraction of the seed extract (eluted with 70% ethanol), as well as kaempferol and five kaempferol derivatives isolated from the seeds showed a moderate inhibitory activity towards α-glucosidase. The fraction had the strongest effect (IC_50_ = 0.62 μg/mL), while kaempferol and its derivatives had IC_50_ values ranging from 6.9 μg/mL to 100 μg/mL [163]. At the dosage of 150 mg/kg/day total flavones from seed residues decreased the fasting plasma insulin level of sucrose-fed rats by 17.79%; human equivalent dose is approximately 24 mg/kg, and required daily intake for a 60 kg person would be 1440 mg [164]. 

Sea buckthorn’s hypoglycemic and anti-inflammatory activity was determined in vivo as well [60,158]. Seed protein (100–200 mg/kg/day) administered by gavage reduced the levels of blood glucose and insulin, and lowered the concentration of pro-inflammatory proteins (IL-6, TNF-α, and NF-κB) in mice with diabetes induced by streptozotocin [60]. 

##### Cytoprotective Activity of Sea Buckthorn Seed Alkaloids (In Vitro)

Two alkaloids from the seeds of sea buckthorn—*N*-*p*-coumaroyl-4-aminobutan-1-ol and hippophamide-protected H9c2 (embryonic rat cardiac) cells from doxorubicin-induced toxicity. They increased cell viability, reduced the levels of ROS, and inhibited the activation of caspase-3 and JNK. The alkaloids inhibited mitochondrial dysfunction as well—they ameliorated the depletion of adenosine triphosphate (ATP) and mitochondrial proteins (Mfn1 and Mfn2), and reduced mtDNA damage [33].

##### Anti-Hyperlipidemic and Anti-Hypercholesterolemic Activity of Sea Buckthorn Seed Oil and Proteins (In Vivo)

Seed protein (100–200 mg/kg/day) administered by gavage had anti-hyperlipidemic and anti-hypercholesterolemic effects. The concentration of total cholesterol, LDL-C, and triglycerides in the serum of diabetic mice were decreased [60]. Moreover, 60-day sea buckthorn seed oil supplementation (1 mL/day) of rabbits fed with high cholesterol diet caused a decrease in the levels of triglycerides and LDL-C, and an increase in the concentration of HDL-C [165].

##### Anti-Hypertensive Activity of Sea Buckthorn Seed Oil and Total Flavones (In Vivo)

Total flavones extracted from sea buckthorn seed residues had an antihypertensive effect on rats fed with high sucrose diet. At the doses of 50, 100, and 150 mg/kg/day they reduced hypertension, and angiotensin II level in the plasma, with results comparable to that of irbesartan at the dose of 20 mg/kg/day. Human equivalent doses in this case are approximately 8, 16, and 24 mg/kg, which means that a 60 kg person would have to ingest 480, 960, or 1440 mg of total flavones from seed residues [164]. In a randomized control trial by Vashishtha et al., supplementation of hypertensive and hypercholesterolemic subjects with 0.75 mL of sea buckthorn oil a day (for 30 days) resulted in decreased systolic and diastolic pressure (by 9.57 and 4.96 mmHg, respectively) and reduction of serum total cholesterol, triglycerides, and LDL cholesterol levels [162]. Moreover, in the beforementioned study by Basu et al., vasorelaxant response of oil-supplemented rabbits fed with normal diet was 35% higher than in control. Vasorelaxant response was decreased in rabbits fed with a high cholesterol diet, but it was brought back to the control level when the animals also received sea buckthorn oil [165].

##### Modulation of Gut Microbiota by Sea Buckthorn Seed Oil and Protein (In Vivo)

Sea buckthorn seed oil could modulate the gut microbiota of hamsters fed with a high cholesterol diet. In experimental groups, 50 or 100% of lard was replaced with the oil. Both doses increased the abundance of *Bacteroidales* family *S24-7*, *Ruminococcaceae*, and *Eubacteriaceae*, at the same time decreasing the number of *Firmicutes.* As a result, the ratio of *Firmicutes* to *Bacteroidetes* was decreased [166]. In another study, Yuan et al. assessed the effect of sea buckthorn seed protein on the profile of intestinal microbes in streptozotocin-induced diabetic mice. The mice received 50, 100, or 200 mg/kg of seed protein. Four-week treatment ameliorated streptozotocin-induced changes in the fecal content of *Bifidobacterium*, *Lactobacillus*, *Bacteroides*, and *Clostridium coccoides* [167].

## 5. Chemical Content of Seeds Eaten Separately and Biological Activity of Seed Extracts, Fractions, Oil, Proteins, Lipids, Polysaccharides, and Isolated Compounds

### 5.1. Gac (Momordica cochinchinensis Sprenger)

#### 5.1.1. Chemical Content of Gac Seeds

*Momordica cochinchinensis*, also known as Gac or red melon, is a tropical plant native to South Asia and northern Australia [56,168]. Although, usually, aril (flesh surrounding the seeds) is the only consumed part of the fruit, in traditional Chinese medicine Gac seeds were considered to have therapeutic effects and were used to treat liver and spleen disorders, fluxes, swelling, bruises, hemorrhoids, or rheumatic pain [56,67]. Lately, studies demonstrated that Gac seeds contain multiple active substances with diverse medicinal properties. They include phenolic compounds (quercetin, luteolin, *p*-hydroxybenzoic acid, gallic acid), terpenoids (Momordica saponin I, lycopene), trypsin inhibitors (MCoTI-I, MCoTI-II, MCoTI-III), and chymotrypsin inhibitors (MCoCI) [42,67,169,170].

#### 5.1.2. Biological Activities of Gac Seed Extracts

##### Antioxidant Activity (In Vitro)

Butanol and ethanol extracts from Gac seeds showed antioxidant activity, measured by DPPH^•+^, ABTS^•+^ and FRAP assays [67]. The values of DPPH^•+^ and FRAP differed between the seed extracts from partially and fully ripe fruits—IC_50_ value of DPPH^•+^ was higher in partially ripe seeds (IC_50_ = 6.66 ± 0.39 mg/mL) than in ripe ones (IC_50_ = 4.20 ± 0.03 mg/mL), while the values of FRAP were higher in ripe (46.49 ± 1.59 μmol FeSO_4_/g) than unripe seeds (36.31 ± 3.42 μmol FeSO_4_/g). These results seem to be correlated with the concentration of phytochemicals. Total phenolic content was higher in the seeds from ripe fruits (2.39 ± 0.26) than in the seeds from partially ripe fruits (1.63 ± 0.22). This was also the case with total flavonoid content (1.88 ± 0.10 vs. 1.57 ± 0.09) [171]. 

##### Neurotrophic Activity (In Vitro)

Seed extract had a neurotrophic effect. It changed the structure, organization, and concentration of neurofilaments, and stimulated the differentiation and elongation of neurites in PC12 cells (at concentration of 150 μg/mL). This effect could be attributed to nerve growth factor (NGF)-mimicking substances contained in the seeds [42]. Further research on this topic ascertained that the protein responsible for these neurogenic effects has a mass of 17 kDa, although more details have yet to be discovered [172]. Small molecule NGF mimetics are investigated as potential therapeutic agents in the treatment of neurodegenerative disorders. Due to their size and structure, they could pass through the brain blood barrier and act similarly to nerve growth factor. Nerve growth factor can facilitate neuronal growth and repair, but stimulating its synthesis comes with a risk of serious adverse effects. For this reason, there is a need for new substances that act similarly to NGF, but have less side effects [42,172].

##### Cytotoxic Activity toward Healthy and Cancer Cells (In Vitro and In Vivo)

Several studies have demonstrated antiproliferative, antimigration, anti-invasion, and cytotoxic effect of *M. cochinchinensis* toward cancer cells [67,168,173]. Seed extract significantly inhibited the proliferation of ZR-75-30 human breast cancer cell line—50% growth inhibitory concentrations were 93.24, 34.04, and 53.43 μg/mL, at the incubation times of 24, 48, and 72 h, respectively. The extract suppressed migration and invasion in a dose-dependent manner. Furthermore, incubation with the concentrations of 30, 60, and 120 μg/mL caused downregulation of matrix metalloproteinase 2 (MMP-2) and matrix metalloproteinase 9 (MMP-9) expression, which may be the reason for the abovementioned anti-metastasis activity [173]. Water extract had a cytotoxic effect on human melanoma cell lines as well. It lowered the viability of D24 cell line by 67%, and C1 by 75%. The 70% ethanol and 50% methanol extracts also showed anticancer activity, but the methanol, 100% ethanol, and butanol extracts did not have such an effect [67].

Unfortunately, recent research has uncovered that Gac seeds can have cytotoxic activity in non-tumorigenic cells as well. Seed extract induced cardiac apoptosis in zebrafish embryos at concentrations ranging from 1.5 to 39.4 ng/fish. It caused cardiac inflammation and increased ROS production, which led to pericardial edema, bradycardia, and reduction of cardiac output. The exact chemical compounds that are responsible for these effects have not been identified yet [170]. In a previous study by Sun et al. mice injected with Gac seed extract experienced adverse effects as well. In this case, saponins were found to be the main compounds responsible for the cytotoxic activity [55].

#### 5.1.3. Biological Activity of Gac Seed Compounds

##### Antioxidant Activity of MCoCI (In Vitro)

MCoCI—a chymotrypsin inhibitor isolated from Gac seeds had antioxidant activity. It protected primary rat hepatocytes from *tert*-butyl hydroperoxide-induced damage. At the concentration of 100 μg/mL MCoCI increased cell viability from 18% to 91%, significantly decreased lipid peroxidation, and restored glutathione concentration to control level. The activity of antioxidant enzymes (glutathione-E-transferase (GST) and SOD) in the cell lysate were increased by 60% and 20%, respectively [174]. 

##### Anti-Inflammatory Activity of Gac Seed Lignans and Saponins (In Vitro)

Lignans isolated from the seeds had anti-inflammatory properties. They inhibited the secretion of NO and TNF-α from RAW 264.7 (murine macrophage) cells stimulated by LPS. IC_50_ value for different compounds varied from 4 μM to 100 μM [169]. Saponins isolated from the seeds (especially gypsogenin 3-*O*-β-d-galactopyranosyl(1→2)-[α-l-rhamnopyranosyl(1→3)]-β-d-glucuronopyranoside) reduced TNF-α-induced inflammation of adipocytes by inhibiting the expression of monocyte chemoattractant protein-1 (MCP-1) and IL-6 [56]. 

### 5.2. Hemp (Cannabis sativa L.)

#### 5.2.1. Chemical Content of Hemp Seeds

In the past, hemp (*Cannabis sativa* L.) has been an important resource, used for food, medicine, oil, and fiber in Europe, Asia, and Africa. Hemp seeds consist of 25–35% lipids, 30% carbohydrates, and 20–25% protein [175]. The oil is a popular product, known for its nutritional value and favorable ratio of *n*-3 to *n*-6 fatty acids (3:1). Hemp seeds contain tocopherols (mostly γ-tocopherol), terpenes, phytosterols, and cannabinoids (e.g., cannabidiolic acid, tetrahydrocannabinolic acid, cannabinol) [175,176,177]. Active substances isolated from the seeds were mostly lignanamides (*N*-caffeoyltyramine, grossamide, cannabisin B, cannabisin F) that constituted 79%, while the content of flavonoids was only 2.8% [37,178,179,180]. 

#### 5.2.2. Biological Activities of Hemp Seeds

##### Anti-Hypercholesterolemic Activity (In Vivo)

Hempseed supplementation (10% feed) of rats fed with high-fat diet (HFD) reduced the amount of total cholesterol, LDL, and triglycerides in plasma, at the same time increasing the levels of HDL. Hemp seeds have also been shown to ameliorate HFD-induced changes in the impedance of aorta walls [181]. 

##### Modulation of Gut Microbiota by Hemp Seeds (In Vivo)

Necib et al. studied the effect of hemp seeds on gut microbiota in mice fed with a high-fat, high sucrose diet. 15% of fat in the diet was substituted with fat from the seeds. This percentage of hemp seed fat in human diet could be attained by daily consumption of 37 g of ‘Finola’ hemp seeds, which is a reasonable amount. The abudance of *Clostridiaceae 1* and *Rikenellaceae* was higher in the group supplemented with hemp seeds, but the Shannon diversity ratio and the proportion of *Firmicutes* to *Bacteroidetes* were not affected [182].

#### 5.2.3. Biological Activities Hemp Seed Proteins, Polysaccharides, Lipids, and Compounds

##### Antioxidant Activity of Hemp Seed Proteins and Polysaccharides (In Vitro and In Vivo)

Numerous studies have noted the antioxidant activity of *C. sativa* [37,58,61,180,183]. Hemp seed protein hydrolysate showed antioxidative activity in vitro. It had good DPPH^•+^ radical, hydroxyl radical, and superoxide radical-scavenging abilities and protected PC12 cells from H_2_O_2_-induced oxidative damage [58]. In an in vivo study by Xue et al., polysaccharides isolated from the seeds could prevent intestinal oxidative damage induced by cyclophosphamide. Supplementation of mice with the doses of 200 mg/kg/day increased the concentration of GPx in serum and ileum of mice, and CAT and SOD in the serum. The rate of lipids peroxidation was decreased as well. This antioxidant activity can be attributed to the upregulation of Nrf2-Kelch-like ECH-associated protein 1 (Keap1) pathway—the polysaccharides increased the expression of Nrf2 and decreased the expression of Keap1 in the ileum [61]. Keap1 is a protein that targets Nrf2 for ubiquitination, which can hamper the antioxidative response of the cell. Human equivalent dose would be approximately 16 mg/kg, 960 mg for a 60 kg person [61,122]. 

##### Anti-Inflammatory Activity of Hemp Seed Lipids and Lignanamides (In Vitro and In Vivo)

Lignanamides from hemp seeds had anti-inflammatory properties in vitro [178,183,184]. Cannabisin F (at the concentration of 15 μM) and grossamide (at the concentration of 20 μM) ameliorated neuroinflammation in BV2 murine microglia cell line stimulated by LPS. NF-κB activation was reduced, and the expression of IL-6 and TNF-α was inhibited [178,184]. The researchers have also discovered that cannabisin F can increase the levels SIRT1. SIRT1 is a histone deacetylase that downregulates NF-κB signaling, and its expression is inhibited by LPS. Thus, it is likely, that anti-inflammatory activity of cannabisin F is caused by prevention of LPS-induced SIRT1 downregulation [184]. Anti-inflammatory activity of hemp was studied in vivo as well—lipid fractions of the seeds that constituted 10% of animals’ feed could decrease the expression of pro-inflammatory enzymes (prostaglandin synthase and thromboxane-A synthase) in rats fed with a high-fat diet [183].

##### Cytotoxic Activity of Hemp Seed Lignanamides toward Cancer Cells (In Vitro)

Evidence shows, that hemp lignanamides possess cytotoxic activity toward cancer cells [37,180]. They significantly lowered the viability of U-87 (human glioblastoma) cell line—at the concentration of 50 μg/mL the change was observable within all the incubation times (24, 48, and 72 h). At the concentration of 25 μg/mL cell viability was decreased after 48 and 72 h incubation. At the same time, no cytotoxic effect was observed on non-tumorigenic human fibroblasts [180]. Cannabisin B inhibited the growth and induced autophagic cell death in HepG2 (human hepatoblastoma) cells. It arrested the cell cycle in the S phase and inhibited survival signaling by suppressing the activation of AKT/mammalian target of rapamycin (mTOR) pathway [37]. 

##### Anti-Hypercholesterolemic and Anti-Hyperlipidemic Activity of Hempseed Lipid Fractions and Oil (In Vivo)

In a study by Kaur et al. hempseed lipid fractions (10% of feed) ameliorated hypercholesterolemia in rats fed with a high-fat diet. HFD-induced renal damage was reduced as well. These effects have been linked to antioxidant and anti-inflammatory properties of hemp seeds. Furthermore, treatment with hempseed lipid fractions had similar efficiency to treatment with statins, which is a promising result that could lead to the development of new strategies for improving the condition of hypercholesterolemic patients [185]. Mokhtari et al. studied the effect of hemp seed oil on hyperlipidemic mice. The doses of 3.5 and 7 mg/kg decreased plasma total cholesterol by 45.7% and 57%, respectively. The concentration of triglycerides was decreased by 82% (at the dose of 3.5 mg/kg) and 87% (at the dose of 7 mg/kg). The levels of LDL-C were decreased by 61%, and the levels of HDL-C increased by 324%. These changes resulted in reduced atherogenic index and LDL/HDL ratio [186]. On the other hand, in a study by Majewski and Jurgoński supplementation of obese male Zucker rats with hemp seed oil (4% diet) decreased the concentration of HDL and triglycerides, but did not affect total cholesterol levels [187].

##### Activity of Cannabidiol (In Vitro and In Vivo)

Cannabidiol (CBD), a well-known constituent of hemp is one of the most extensively studied cannabinoids. In 2018, it has been approved for the treatment of seizures occurring in two types of pediatric epilepsy—Lennox-Gastaut syndrome and Dravet syndrome [36,188]. It shows several potential health benefits observed in experimental models of cardiovascular diseases (antioxidant, anti-inflammatory, vasorelaxant, protection against reperfusion injury), neurological diseases (anti-apoptotic, anti-convulsant, anxiolytic, antidepressant, amyloid β clearance), and cancer (activity against lung cancer, colon cancer, breast cancer, prostate cancer, brain cancer, and melanoma) [36,188,189]. Despite the promising evidence for numerous beneficial properties of CBD in vitro, there is still a need for more clinical trials that could ascertain if the use of cannabidiol can bring the expected effects in vivo as well, and confirm its safety [57,190]. Moreover, the content of CBD in the seeds is lower than in inflorescences and leaves. In ‘Finola’ variety the concentration of CBD in the seeds was 14.80 ± 0.32 mg/kg. Leaves had slightly more CBD (16.54 ± 1.20 mg/kg), but the highest concentration was recorded in female plants’ inflorescences (135.41 ± 8.64 mg/kg). Despite this Pexová Kalinová et al. suggests that seed hulls could still be used for the extraction of cannabidiol, cannabidiolic, and canabigerolic acids, though in comparison to leaves and inflorescences the yield was lower by approximately 5.6 g/ha [191].

### 5.3. Sacha inchi (Plukenetia volubilis L.) 

#### 5.3.1. Chemical Content of Sacha Inchi Seeds

Sacha inchi (*Plukenetia volubilis* L.), also called Inca peanut, is a plant from the *Euphorbiaceae* family, native to tropical rainforests of South America [16,192,193]. Its seeds are encapsulated in star-shaped fruits, and contain tocopherols (mostly γ-tocopherol, δ-tocopherol), phytosterols (β-sitosterol, stigmasterol, campesterol), flavonoids, lignans, phenyl alcohols, and secoiridoids, although, more research is needed to ascertain their exact content [192,194]. The proportion of omega-6 (ω-6) to omega-3 (ω-3) fatty acids is close to one, which is the optimal ratio [195,196].

#### 5.3.2. Biological Activities of Sacha Inchi Seed Extracts

##### Antioxidant Activity (In Vitro)

Seed extract had radical-scavenging capacity. It could reduce the ABTS^•+^ cation (0.49 μmol TE/g seed). Interestingly, roasting the seeds improved their antioxidant activity against DPPH^•+^ (an 13% increase) and peroxyl radicals (an 22% increase) [20].

##### Cytotoxic Activity (In Vitro)

Sacha inchi seeds were mildly cytotoxic toward human hepatic stellate cells (LX-2), with the IC_50_ value of 500 μg/mL, but this effect could be eliminated by roasting [197]. 

#### 5.3.3. Biological Activities of Sacha Inchi Seed Polysaccharide, Proteins, and Oil

##### Antioxidant Activity of Sacha Inchi Seed Polysaccharide and Oil (In Vitro and In Vivo)

PVLP-1, a polysaccharide extracted from the seeds showed strong radical-scavenging activity against ABTS^•+^, DPPH^•+^, hydroxyl, and superoxide anion radicals [16]. Sacha inchi oil, which is rich in alpha-linoleic acid, improved oxidative stress parameters in the liver of rats. Inclusion of the oil in the diet (10% feed) raised the concentration of hepatic GSH and increased the activity of SOD, CAT, GPX, and glutathione reductase (GR) [198].

##### Immunomodulatory Activity of Sacha Inchi Seed Polysaccharide and Protein (In Vitro)

Apart from its antioxidant activity, PVLP-1 could improve proliferation and viability of RAW 264.7 cells as well. It also upregulated the secretion of TNF-α, IL-1β, IL-6, and nitric oxide [16]. Albumin fraction of sacha inchi seed protein had similar effects as PVLP-1. It stimulated the proliferation of mouse spleen lymphocytes. The proliferation rates of cells treated with 10 and 20 μg/mL of albumins were higher than positive control (cells stimulated with LPS or Concanavalin A). The albumins increased levels of TNF-α as well [193].

##### Anti-Hypercholesterolemic, Antihyperlipidemic, and Anti-Hypertensive Activity of Sacha Inchi Seed Oil (In Vivo)

In a clinical trial that aimed to determine the effect of sacha inchi seed oil on postprandial lipid profile, healthy subjects consuming high-fat meals with the addition of seed oil (15 mL/day) had lower serum levels of total cholesterol than subjects that consumed high-fat meals without oil supplementation. The levels of triglycerides and HDL-C remained unaffected [199]. In another clinical trial, the concentration of serum total cholesterol and LDL-C decreased as well. Furthermore, systolic and diastolic blood pressure was lower than before oil supplementation at the dose of 10–15 mL/day over the span of four months [200].

##### Anti-Inflammatory Activity of Sacha Inchi Proteins and Oil (In Vitro and In Vivo)

Quinteros et al. have reported, that sacha inchi proteins had anti-inflammatory properties. At the concentrations of 100–1000 μg/mL, they inhibited albumin denaturation by 7.77 ± 0.3%–78.13 ± 0.44%. At the highest concentration, their activity was comparable to that of diclofenac sodium (200 μg/mL), which is an anti-inflammatory drug [201]. In vivo protein denaturation is one of the causes of inflammation, and can lead to arthritis and other inflammatory diseases. For this reason, denaturation-preventing agents are valued as potential candidates for anti-inflammatory drugs [202]. Anti-inflammatory activity of sacha inchi was studied in vivo as well. In rats with hyperuricemia, administration of 0.42 g/kg of seed press-cake protein hydrolysate decreased the expression of pro-inflammatory genes (*Tlr4*, *Map3k8*, *Ikbke*, *Nlrp3*, *Pik3cg*, *Pik3ap1*, and *Akt3*) in the kidneys [203]. Moreover, in a clinical trial by Alayón et al. sacha inchi oil reduced the levels of postprandial inflammation induced by eating a high-fat breakfast. 15 mL of seed oil ingested with the meal, decreased the concentration of IL-6 in the serum of metabolically healthy participants [199].

##### Modulation of Gut Microbiota by Sacha Inchi Oil and Press-Cake Protein (In Vivo)

In a hyperuricemic rat model, supplementation of 0.42 g/kg body weight (BW) of seed press-cake protein hydrolysate improved the diversity of gut microbiota. Moreover, it increased the abundance of probiotic bacteria like *Akkermansia* and *Alistipes*, at the same time reducing the *Streptococcus* population [203]. Sacha inchi oil had similar effects. A dose of 1.5 mL/kg reversed gut microbiota dysbiosis in rats fed with high-fat diet. The number of *Roseburia*, *Turicibacter*, and *Butyrivibrio* increased, while the levels of *Enterobacteriaceae*, *Escherichia*, and *Bacteroides* were reduced [204].

The biological activities of seed extracts, fractions, and oil were compiled in Figure 2. 

Active compounds isolated from the seeds included in this review, as well as their biological activity are compiled in Table 1 and Table 2. 

## 6. Conclusions

Seeds contain many chemical compounds that possess diverse biological activities, including antioxidant, anti-inflammatory, antibacterial, anti-apoptotic, anticoagulant, antihypertensive, hypocholesterolemic, or antiproliferative. Despite the promising results obtained in in vitro experiments we still do not have enough results from animal studies and clinical trials. Among the factors that determine in vivo effectiveness of extracts or chemical substances, bioavailability is of great importance. Some phytochemicals, like anthocyanins, are known for their poor bioavailability. This can pose a problem that would have to be resolved if we were to consider implementing these substances in medical scenarios [32,304]. Fortunately, there are ways to improve bioavailability of a compound—for example by converting it into smaller molecules, combining it with other compounds, or enclosing it into lipid vesicles, polymeric nanoparticles or other types of nanomaterials [69,74,75,76]. Assessing the safety of studied extracts, oils, or compounds is immensely important as well [3,51,170]. 

In the future, more emphasis should be put on animal studies and clinical trials. This step is needed to further our knowledge of the preparations and compounds that have well documented in vitro activity, and check if they can be used as functional foods, nutraceuticals, or pharmaceuticals. It must be noted, that oftentimes the concentration of active compounds in seeds is too low to be effective only through dietary consumption. When this is the case, using the extracts, fractions, oil, proteins, polysaccharides, or compounds obtained from seeds in pharmaceutical products is a more realistic option.

In conclusion, seeds are a source of many promising compounds that have the potential to be implemented in the prevention or treatment of diseases in the future, but the process of introducing them in conventional medicine must be preceded by a thorough in vivo investigation of their effectiveness and safety.

## Figures and Tables

**Figure 1 nutrients-15-00187-f001:**
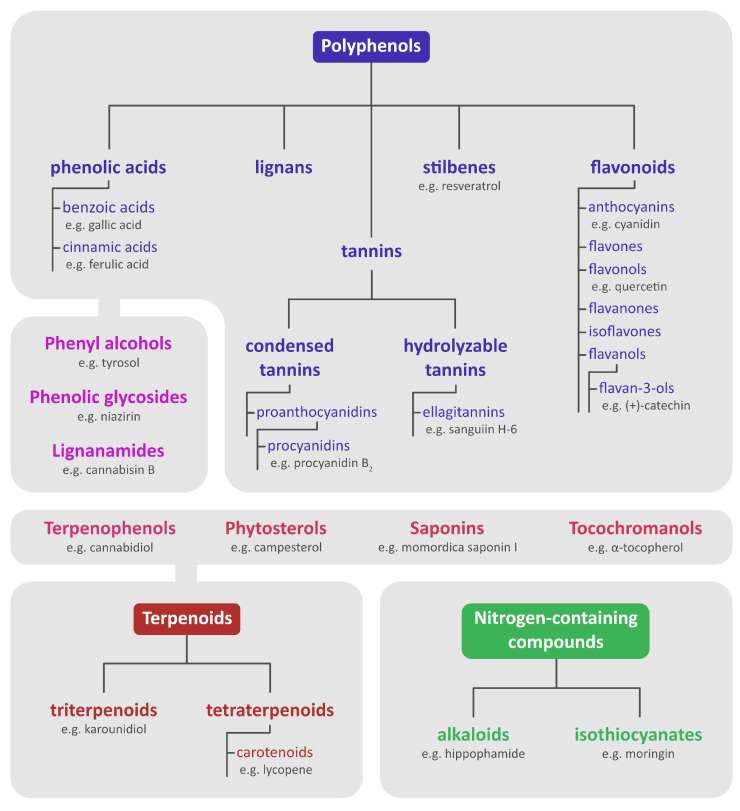
Classification of phytochemicals. The figure includes only the compounds that were mentioned in the text. Cojoined grey fields indicate a certain degree of structural similarity, or involvement of one group of phytochemicals or their precursors in the synthesis of the other. Compilation of data: [4,23,28,29,30,31,32,33,34,35,36,37,38,39,40,41].

**Figure 2 nutrients-15-00187-f002:**
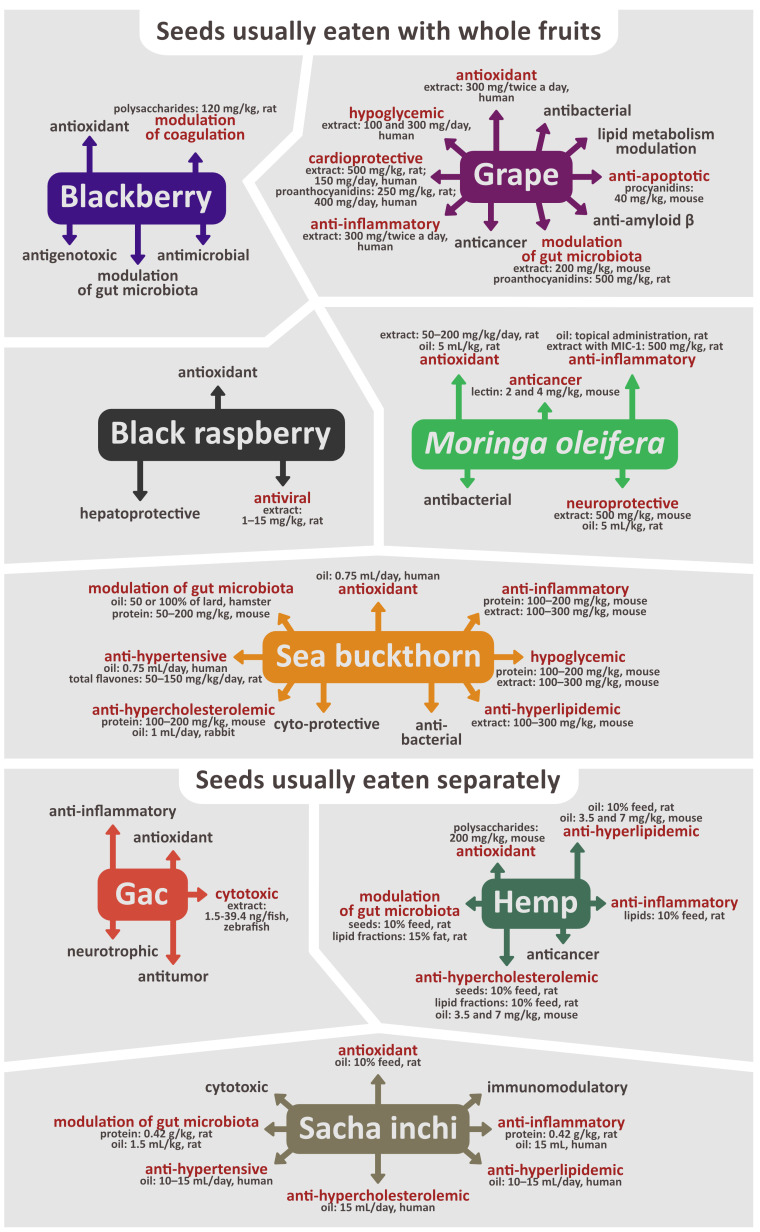
Biological activity of seed extracts, fractions, and oil. The activity of singular isolated compounds was not included, as it is presented in Table 1. The activities that were studied in vivo are marked with red font. The figure is a compilation of data from Section 4 and Section 5. MIC-1, moringa isothiocyanate-1.

**Table 1 nutrients-15-00187-t001:** Compounds isolated from seeds usually eaten with whole fruits and their biological activity.

Type of Compound	Compound	Plant Species	Biological Activity	Citations
Phenolic acids	Protocatechuic acid	Blackberry Black raspberry Grape Sea buckthorn	Antioxidant **, anti-inflammatory **, antibacterial, antiviral **, antifibrotic, anti-atherosclerotic, anti-amyloid β **, anti-hypertensive **, decreasing insulin resistance **, hypolipidemic **, hypoglycemic ** Anti-aging, antidiabetic **, anticancer **, antiosteoporosis **; cardioprotective **, neuroprotective **	[6,28,99,156,205,206]
	p-Coumaric acid	Blackberry Black raspberry Grape	Antioxidant **, anti-inflammatory **, antimicrobial, antiviral, antiplatelet **, antiproliferative **, decreasing insulin resistance ** Anticancer **, antidiabetic **	[6,99,207,208]
	Gallic acid *	Blackberry Black raspberry Grape M. oleifera	Antioxidant **, prooxidant, anti-inflammatory **, antimicrobial, antiviral, hypoglycemic **, hypolipidemic ** Anticancer **, anti-obesity **, cardioprotective **, gastroprotective **, hepatoprotective **, neuroprotective **	[6,7,67,92,109,209]
	Syringic acid	Blackberry Black raspberry Grape	Antioxidant **, anti-inflammatory **, antimicrobial, anti-angiogenic, antiendotoxic ** Anticancer **, antidiabetic **, cardioprotective **, hepatoprotective **, neuroprotective **	[6,210,211,212]
	Caffeic acid *	Blackberry Black raspberry Grape M. oleifera	Antioxidant **, prooxidant, anti-inflammatory **, modulation of gut microbiota **, suppression of metalloproteinases expression Anticancer **, hepatoprotective **	[6,7,92,98,213,214]
	Caftaric acid	Grape	Antioxidant **, anti-apoptotic, antimutagenic, antidiuretic **, anti-hypercholesterolemic ** Anticancer, hepatoprotective **, nephroprotective **	[1,215,216,217]
	Vanillic acid	Grape	Antioxidant **, anti-inflammatory **, antibacterial, antidepressant **, antihypertensive ** Anti-asthma **, antidiabetic, cardioprotective **, nephroprotective **, neuroprotective **	[99,218,219,220,221,222,223,224,225]
	Chlorogenic acid	Grape Sea buckthorn	Antioxidant **, anti-inflammatory **, immunomodulatory **, antibacterial, antiviral, anti-hypercholesterolemic **, anti-hyperlipidemic **, hypoglycemic **, antimutagenic Anti-cancer, anti-obesity **, cardioprotective **, hepatoprotective **	[1,156,216,226]
	Ferulic acid	Black raspberry Grape	Antioxidant **, anti-inflammatory **, antiplatelet **, anti-apoptotic **, antifibrosis **	[92,99,170,227]
			Cardioprotective **	
	Ellagic acid *	Blackberry Black raspberry M. oleifera	Antioxidant **, anti-inflammatory **, antibacterial, anti-viral **, anti-apoptotic, antidepressant, anti-atherosclerotic **, antihyperglycemic **, antihyperlipidemic **, anti-angiogenic ** Antidiabetic **, anticancer **, cardioprotective **, hepatoprotective, nephroprotective **, neuroprotective **	[3,7,82,87,92]
Flavonoids	Catechins *	Blackberry Black raspberry Grape M. oleifera Sea buckthorn	Antioxidant **, anti-inflammatory **, antibacterial **, antiviral, anti-apoptosis ** antihypertensive **, protecting from UV radiation ** Anticancer **, anti-obesity **, cardioprotective **, hepatoprotective **	[6,31,109,156,228,229,230,231,232,233,234,235]
	Quercetin *	Blackberry Black raspberry Grape M. oleifera Sea buckthorn	Antioxidant **, prooxidant, anti-inflammatory **, antimicrobial, anti-hypertensive **, hypoglycemic **, anti-hyperlipidemic **, anti-hypercholesterolemic ** Anticancer **, anti-obesity **, anti-diabetes **, cardioprotective **	[5,6,7,8,92,170,180,233,236,237]
	Myricetin	Blackberry Sea buckthorn	Antioxidant **, prooxidant **, anti-inflammatory **, antiviral, anti-atherosclerotic **, antiplatelet ** Anticancer **, cardioprotective **, neuroprotective	[6,156,238,239,240]
	Kaempferol *	Blackberry Grape M. oleifera Sea buckthorn	Antioxidant **, anti-inflammatory **, antithrombotic **, antihyperglycemic **, antihyperlipidemic **, anti-atherosclerotic ** Anticancer **, antidiabetic **, anti-obesity **, cardioprotective **, gastroprotective **, hepatoprotective **, neuroprotective **, osteoprotective **	[6,7,11,98,154,180,241,242,243,244,245,246]
	Rutin	Black raspberry Grape Sea buckthorn	Antioxidant **, anti-inflammatory **, antibacterial, antihyperglycemic **, anti-hypercholesterolemic **, antihypertensive **, antiplatelet, antiulcer ** Anti-asthmatic **, anti-arthritic **, anticancer **, anticataract **, antidiabetic **, anti-osteoporotic **, cardioprotective **, hepatoprotective **, nephroprotective **, neuroprotective **, wound healing **	[5,76,92,156,170,247,248]
	Isorhamnetin *	Sea buckthorn	Antioxidant **, anti-inflammatory **, anti-apoptotic **, antibacterial, antiviral **, anti-atherogenic **, anticholinesterase, anticoagulant, antihypertensive, antifibrosis **, hypoglycemic ** Anticancer **, anti-osteoporosis **, anti-obesity **, cardioprotective **, hepatoprotective **, nephroprotective **, neuroprotective **	[8,249,250]
	Luteolin *	Grape	Antioxidant **, anti-inflammatory **, antidepressant **, hypoglycemic **, improving insulin sensitivity ** Anticancer **, anti-obesity **, cardioprotective **, hepatoprotective **, neuroprotective **	[1,70,170]
Anthocyanins	Peonidin-3-O-glucoside	Blackberry Black raspberry	Stimulating bone formation ** Anticancer	[6,251,252]
	Cyanidin-3-glucoside	Black raspberry Grape	Antioxidant **, anti-inflammatory **, antimicrobial, antiviral, antiatherogenic **, antithrombotic, hypoglycemic **, decreasing insulin resistance ** Anticancer **, antidiabetic **, anti-obesity **, cardioprotective **, neuroprotective **	[5,6,96,253]
	Cyanidin-3-rutinoside	Black raspberry	Antioxidant, prooxidant, proapoptotic (in cancer cells), hypoglycemic ** Anticancer, antidiabetic	[30,92,254,255]
Procyanidins	Procyanidin B_1_	Blackberry Black raspberry	Anti-inflammatory, antiviral Anticancer	[6,256,257,258]
	Procyanidin B_2_	Grape	Antioxidant **, anti-inflammatory **, anti-apoptotic, proapoptotic **, anti-atherogenic, anti-angiogenic **, pro-angiogenic **, improving endothelial function, improving mitochondrial function, anti-fibrosis ** Anticancer **, hepatoprotective **, wound healing **	[53,259,260,261,262,263,264,265]
Stilbenes	Resveratrol	Black raspberry Grape	Antioxidant **, anti-inflammatory **, antimicrobial, anti-apoptotic **, antiglycation **, antiplatelet, increased NO synthesis **, stimulating bone formation ** Anti-ageing **, anticancer **, antidiabetic **, cardioprotective **, neuroprotective **	[5,74,92,101,266]
Phenyl alcohols	Tyrosol	Grape	Antioxidant **, anti-inflammatory **, antibacterial, anti-apoptotic, antiplatelet, antihyperlipidemic **, hypoglycemic **, anti-genotoxic ** Anti-obesity, cardioprotective **, anti-diabetes **, neuroprotective **	[38,99,267,268,269,270,271]
Phenolic glycosides	Niazirin	M. oleifera	Antioxidant **, anti-inflammatory **, antiproliferative **, hypoglycemic **, anti-hyperlipidemic **, decreasing insulin resistance ** Antidiabetic **	[39,150,151]
Phytosterols	Campesterol	Blackberry M. oleifera	Prooxidant (in cancer cells), anti-neuroinflammatory, hypocholesterolemic Anticancer	[7,79,192,272,273,274]
	Stigmasterol	Blackberry M. oleifera	Antioxidant, prooxidant, anti-inflammatory **, anti-apoptotic **, anti-hypercholesterolemic **, anti-hyperlipidemic **, anti-nociceptive ** Antiallergic **, anti-asthmatic **, anti-arthritis **, anticancer **, antidiabetic **, neuroprotective **	[7,79,192,272,273,275,276,277,278,279,280,281,282]
	β-sitosterol	Blackberry M. oleifera Sea buckthorn	Antioxidant **, anti-inflammatory **, antibacterial **, anti-acetylcholinesterase, anti-amyloid β, anti-hypercholesterolemic **, anti-hyperlipidemic **, anxiolytic **, immunomodulatory **, modulation of gut microbiota ** Anticancer **, antidiabetic **, cardioprotective **, hepatoprotective **, neuroprotective **	[7,79,155,192,272,273,281,283,284,285,286,287]
Tocopherols	α-tocopherol	M. oleifera Sea buckthorn	Antioxidant **, anti-inflammatory **, antiproliferative **, anti-hypercholesterolemic **Anticancer **, cardioprotective **	[7,34,175,288,289,290,291]
	γ-tocopherol	Blackberry M. oleifera Sea buckthorn	Antioxidant **, anti-inflammatory ** Anti-asthmatic **, antiallergic **, anticancer **	[7,34,79,175,196,288,292]
	δ-tocopherol	Blackberry M. oleifera	Antioxidant, anti-inflammatory, antiproliferative **, proapoptotic ** Anticancer **	[7,79,175,196,288,293,294]
Ellagitannins	Sanguiin H-6	Blackberry Black raspberry	Antioxidant **, antibacterial **, anti-apoptotic **, anti-angiogenic, anti-osteoclastogenic **, anti-genotoxic, estrogenic Anticancer, anti-osteoporosis **	[2,48,49,50,51,52,82,89,92]
Isothiocyanates	Moringa isothiocyanate-1 (moringin)	M. oleifera	Antioxidant, anti-inflammatory **, pro-apoptotic (in cancer cells), antinociceptive ** Antidiabetic **, neuroprotective, pain-relieving **	[31,39,130,145,146,147,295,296]
Alkaloids	N-p-coumaroyl-4-aminobutan-1-ol	Sea buckthorn	Antioxidant, cytoprotective	[33]
	Hippophamide	Sea buckthorn	Antioxidant, cytoprotective	[33]

* and/or its derivatives; ** the activity was studied in vivo. UV, ultraviolet; NO, nitric oxide.

**Table 2 nutrients-15-00187-t002:** Compounds isolated from seeds usually eaten separately from fruits and their biological activity.

Type of Compound	Compound	Plant Species	Biological Activity	Citations
Phenolic acids	Gallic acid *	Gac	Antioxidant **, prooxidant, anti-inflammatory **, antimicrobial, antiviral, hypoglycemic **, hypolipidemic ** Anticancer **, anti-obesity **, cardioprotective **, gastroprotective **, hepatoprotective **, neuroprotective **	[6,7,67,92,109,209]
	Ferulic acid	Gac	Antioxidant **, anti-inflammatory **, antiplatelet **, anti-apoptotic **, antifibrosis ** Cardioprotective **	[92,99,170,227]
Flavonoids	Quercetin *	Gac Hemp	Antioxidant **, prooxidant, anti-inflammatory **, antimicrobial, anti-hypertensive **, hypoglycemic **, anti-hyperlipidemic **, anti-hypercholesterolemic ** Anticancer **, anti-obesity **, anti-diabetes **, cardioprotective **	[5,6,7,8,92,170,180,233,236,237]
	Kaempferol *	Hemp	Antioxidant **, anti-inflammatory **, antithrombotic **, antihyperglycemic **, antihyperlipidemic **, anti-atherosclerotic ** Anticancer **, antidiabetic **, anti-obesity **, cardioprotective **, gastroprotective **, hepatoprotective **, neuroprotective **, osteoprotective **	[6,7,11,98,154,180,241,242,243,244,245,246]
	Rutin	Gac	Antioxidant **, anti-inflammatory **, antibacterial, antihyperglycemic **, anti-hypercholesterolemic **, antihypertensive **, antiplatelet, antiulcer ** Anti-asthmatic **, anti-arthritic **, anticancer **, anticataract **, antidiabetic **, anti-osteoporotic **, cardioprotective **, hepatoprotective **, nephroprotective **, neuroprotective **, wound healing **	[5,76,92,156,170,247,248]
	Luteolin *	Gac	Antioxidant **, anti-inflammatory **, antidepressant **, hypoglycemic **, improving insulin sensitivity ** Anticancer **, anti-obesity **, cardioprotective **, hepatoprotective **, neuroprotective **	[1,70,170]
Phytosterols	α-spinasterol	Gac	Anti-inflammatory **, antinociceptive **, antiseizure **, antiulcer, antiproliferative Anticancer, pain-relieving **	[169,297,298,299]
	Campesterol	Sacha inchi	Prooxidant (in cancer cells), anti-neuroinflammatory, hypocholesterolemic Anticancer	[7,79,192,272,273,274]
	Stigmasterol	Sacha inchi	Antioxidant, prooxidant, anti-inflammatory **, anti-apoptotic **, anti-hypercholesterolemic **, anti-hyperlipidemic **, anti-nociceptive ** Antiallergic **, anti-asthmatic **, anti-arthritis **, anticancer **, antidiabetic **, neuroprotective **	[7,79,192,272,273,275,276,277,278,279,280,281,282]
	β-sitosterol	Sacha inchi	Antioxidant **, anti-inflammatory **, antibacterial **, anti-acetylcholinesterase, anti-amyloid β, anti-hypercholesterolemic **, anti-hyperlipidemic **, anxiolytic **, immunomodulatory **, modulation of gut microbiota ** Anticancer **, antidiabetic **, cardioprotective **, hepatoprotective **, neuroprotective **	[7,79,155,192,272,273,281,283,284,285,286,287]
Terpenoids	Lycopene	Gac	Antioxidant **, anti-inflammatory **, anti-atherogenic **, antiplatelet **, antihypertensive **, anti-hyperlipidemic **, anti-seizure **, proapoptotic (in cancer cells) ** Anticancer **, anti-obesity **, cardioprotective **, neuroprotective **	[42,43,44,45,46]
Terpenophenols	Cannabidiol	Hemp	Antioxidant **, anti-inflammatory **, anti-apoptotic **, proapoptotic **, anti-angiogenic **, antiseizure **, anti-amyloid β **, antidepressant **, antipsychotic **, anxiolytic ***,* antiarthritic **, immunomodulatory **, vasodilatory ** Anticancer **, cardioprotective **, neuroprotective **	[36,188,189,300,301]
Tocopherols	α-tocopherol	Hemp	Antioxidant **, anti-inflammatory **, antiproliferative **, anti-hypercholesterolemic **Anticancer **, cardioprotective **	[7,34,175,288,289,290,291]
	γ-tocopherol	Hemp Sacha inchi	Antioxidant **, anti-inflammatory ** Anti-asthmatic **, antiallergic **, anticancer **	[7,34,79,175,196,288,292]
	δ-tocopherol	Hemp Sacha inchi	Antioxidant, anti-inflammatory, antiproliferative **, proapoptotic ** Anticancer **	[7,79,175,196,288,293,294]
Lignanamides	*N*-caffeoyltyramine *	Hemp	Antioxidant, anti-inflammatory, antiproliferative	[179,302,303]
	Grossamide	Hemp	Anti-neuroinflammatory Neuroprotective	[178]
	Cannabisin B	Hemp	Autophagic, antiproliferative	[37,180]
	Cannabisin F	Hemp	Anti-neuroinflammatory Neuroprotective	[184]

* and/or its derivatives; ** the activity was studied in vivo.

## Data Availability

Not applicable.

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
