# Peer review of "Selected Seeds as Sources of Bioactive Compounds with Diverse Biological Activities"

_nutrients, 2022, doi:10.3390/nu15010187_

Round 1
Reviewer 1 Report
Minor comments:
1. Title: The title needs revision and can be made more attractive.
2. Abstract: The abstract is written well. However, elaboration from which databases the literature was collected.
3. Section 1: Introduction: The introduction is very concise. The background and rationale of the study are written very well written.
Though authors have added a section on how literature was collected however how data was retrieved and extracted, how many papers were browsed, excluded paper and included paper flow chart need to be added, and which software was used for retrieval of data. A rearrangement of the sections is required to understand the manuscript.
4. The authors may add a paragraph on discussion, gaps and future recommendation
5. References. In line 927 reference in Harvard method. Kindly check the whole document for referencing

Author Response
Thank you for reviewing the manuscript and providing such helpful comments. All of them have been taken into consideration when revising the manuscript.
Minor comments:
- Title: The title needs revision and can be made more attractive.
Response: We have changed the title. Now, it is” Selected Seeds as Sources of Bioactive Compounds with Diverse Biological Activities”
- Abstract:The abstract is written well. However, elaboration from which databases the literature was collected.
Response: We have added this information: ”This review is based on studies identified in electronic databases, including PubMed, ScienceDirect, and SCOPUS.”
- Section 1: Introduction: The introduction is very concise. The background and rationale of the study are written very well written.
Though authors have added a section on how literature was collected however how data was retrieved and extracted, how many papers were browsed, excluded paper and included paper flow chart need to be added, and which software was used for retrieval of data. A rearrangement of the sections is required to understand the manuscript.
Response: We have added some more exclusion criteria: “articles about plant parts other than seeds, articles about the effects of seeds combined with other plant parts (e.g. skin or pulp), and articles about a combination of extracts or compounds from different plant species.”, however our review is not a systematic review, so we did not collect the data about the exact number of collected and excluded papers. The manuscript was rearranged (seeds were divided into two sections – seeds usually eaten with whole fruits and seeds usually eaten separately). Moreover, we have rearranged the sections.
- The authors may add a paragraph on discussion, gaps and future recommendation.
Response: We have revised the conclusions section and added a paragraph on discussion, gaps in knowledge and future recommendations. Moreover, we have added a summary of the biological activity of all seed extracts, oil, proteins, polysaccharides, and compounds included in the text:
“Blackberry seed extracts showed antioxidant, antimicrobial, and antigenotoxic activity. Seed flour could change the profile of gut microbiota, and polysaccharides modulated coagulation. Ellagic acid and sanguiin H-6 are prominent constituents of the seeds. Ellagic acid has well-documented anti-inflammatory activity, while sanguiin H-6 displays a wide array of biological activities, including antioxidant, antibacterial, antitumor, anti-angiogenic, estrogenic, and anti-osteoclastogenic.”
“Extracts from black raspberry seeds displayed antioxidant and antiviral activity. Polyphenols isolated from the seeds (gallic acid and cyanidin-3-glucoside) also inhibited viral infection, while seed oil had hepatoprotective properties in vitro.”
“Grape seed extracts had antioxidant, anti-inflammatory, antitumor, cardioprotective, hypoglycemic, and antibacterial activity. Moreover, they inhibited the oligomerization of amyloid β, modulated lipid metabolism, and facilitated the recovery of gut microbiota after treatment with antibiotics. Seed oil had antioxidant, anti-inflammatory, and hypoglycemic properties as well. Various compounds isolated from grape seeds (including proanthocyanidins and procyanidins) showed antioxidant, anti-apoptotic, antitumor, and cardioprotective activities, and modulated gut microbiota.”
“Extracts from the seeds of Moringa oleifera had antioxidant, anti-inflammatory, neuroprotective, cytotoxic, and antibacterial effects. Moringa seed oil showed anti-inflammatory, neuroprotective, and cytotoxic activities as well. Seed proteins were effective antioxidants, while seed lectin inhibited the growth of EAC carcinoma cells and had antibacterial properties. MIC-1 and niazirin had promising biological effects as well. MIC-1 had anti-inflammatory, antioxidant, pain-relieving, and antitumor activity. Niazirin showed hypoglycemic, antioxidant, anti-inflammatory, and anti-hypercholesterolemic effects.”
“Sea buckthorn seed extracts had antioxidant, hypoglycemic, anti-inflammatory, anti-hyperlipidemic, anti-hypercholesterolemic, and antibacterial effects. Moreover, seed oil showed antioxidant, antihypertensive, anti-hypercholesterolemic, and anti-hyperlipidemic activity. Seed proteins had anti-inflammatory, anti-hypercholesterolemic, and anti-hyperlipidemic properties, while flavonoids could reduce oxidative stress and inflammation. Both seed protein and oil could modulate gut microbiota as well. Total flavones from sea buckthorn seeds showed antihypertensive activity, while alkaloids protected embryonic rat cardiac cells from doxorubicin-induced toxicity.”
“Gac seed extracts had antioxidant, neurotrophic, antitumor, and cytotoxic effects. MCoCI (a chymotrypsin inhibitor) isolated from Gac seeds had antioxidant activity as well. Lignans and saponins had anti-inflammatory properties.”
“Hemp seeds and seed lipid fractions could decrease the levels of cholesterol in rats fed with high-fat diet. Seed proteins and polysaccharides showed antioxidant activity, while lignanamides reduced inflammation and had antitumor effects. Cannabidiol, a well-known constituent of hemp has been shown to possess many beneficial properties, including anti-seizure, cardioprotective, antioxidant, anti-inflammatory, antidepressant, and anticancer. Moreover, hemp seeds altered the content of of Clostridiaceae 1 and Rikenellaceae in the gut.”
“Sacha inchi seed extract had antioxidant activity. The seeds were also cytotoxic toward healthy human cells, but this could be attenuated through roasting. Seed oil had antioxidant activity as well. It also showed anti-hypercholesterolemic, antihyperlipidemic, anti-hypertensive, anti-inflammatory effects. Seed polysaccharide (PVLP-1) and protein has immunomodulatory activity. Moreover, proteins reduced inflammation and modulated gut microbiota, while polysaccharide had antioxidant properties.”
“Despite the promising results obtained in in vitro experiments we still do not have enough results from animal studies and clinical trials. Among the factors that determine in vivo effectiveness of extracts or chemical substances, bioavailability is of great importance. Some phytochemicals, like anthocyanins, are known for their poor bioavailability. This can pose a problem that would have to be resolved if we were to consider implementing these substances in medical scenarios [28,292]. Fortunately, there are ways to improve bioavailability of a compound – for example by converting it into smaller molecules, combining it with other compounds, or enclosing it into lipid vesicles, polymeric nanoparticles or other types of nanomaterials [46,51–53]. Assessing the safety of studied extracts, oils, or compounds is immensely important as well [3,68,152].”
“In the future, more emphasis should be put on animal studies and clinical trials. This step is needed to further our knowledge of the preparations and compounds that have well documented in vitro activity, and check if they can be used as functional foods, nutraceuticals, or pharmaceuticals. It must be noted, that oftentimes the concentration of active compounds in seeds is too low to be effective only through dietary consumption. When this is the case, using the extracts, fractions, oil, proteins, polysaccharides, or compounds obtained from seeds in pharmaceutical products is a more realistic option.“
- In line 927 reference in Harvard method. Kindly check the whole document for referencing
Response: Reference was corrected; all references were checked.
Reviewer 2 Report
Manuscript 2117671
Title: Selected Seeds as Sources of Compounds with Diverse Biological Activities
Journal Nutrients
The review entitled “Selected Seeds as Sources of Compounds with Diverse Biological Activities” summarizes the biological activities of selected seeds (blackberry, black raspberry, grape, Moringa oleifera Lam., sea buckthorn, Gac, hemp, and sacha inchi). Different biological activities refer to extracts, fractions, oil, flour, proteins, polysaccharides, or purified chemical compounds isolated from the seeds. The review is interesting but several parts should be better described, additional information is necessary in some parts, and a general re-organization is suggested. Please follow the comments in the file.

Author Response
The review entitled “Selected Seeds as Sources of Compounds with Diverse Biological Activities” summarizes the biological activities of selected seeds (blackberry, black raspberry, grape, Moringa oleifera Lam., sea buckthorn, Gac, hemp, and sacha inchi). Different biological activities refer to extracts, fractions, oil, flour, proteins, polysaccharides, or purified chemical compounds isolated from the seeds. The review is interesting but several parts should be better described, additional information is necessary in some parts, and a general re-organization is suggested. Please follow the comments in the file.
Thank you for reviewing the manuscript and providing such helpful comments. All of them have been taken into consideration when revising the manuscript.
L34-44 Please enrich this part including the extraction of seed proteins, polysaccharides, fractions, and purified chemical compounds. Please be more specific with selected examples.
Response: We have added information about extraction of proteins and polysaccharides: ‘There are many approaches to extracting different components from seeds. The methods used to extract proteins include alkali extraction, ultrasonic assisted alkali extraction, extraction by fractionation method, enzymatic extraction, or enzyme-assisted micro-fluidization. For example, González Garza et al. (2017) used alkaline extraction with isoelectric precipitation to obtain bioactive peptides from Moringa oleifera seeds [14,15]. Polysaccharides can be obtained by hot water extraction, ultrasound-assisted extraction, ultrasound-microwave-assisted extraction, ultrahigh pressure-assisted extraction, enzyme-assisted extraction, subcritical water extraction, or pulsed electric field assisted extraction. For example, Tian et al. (2020) extracted polysaccharides from sacha inchi seed powder with the help of hot water reflux method [16,17].’ We have added more information about extraction of purified compounds later in the text.
L45-57 Which are the most important nutrients in dietary seeds? Please discuss this aspect and revise the text
Response: We have added new information about nutrients in the seeds: “Some of the most important nutrients contained in ‘specialty seeds’ (high value and/or uncommon seeds) include polyunsaturated fatty acids, dietary fiber, minerals (like potassium, phosphorus, calcium, and magnesium, vitamins, and amino acids [20].”
L53-57 What about the presence of antinutritional factors? Please add relevant references related to the presence of antinutritional factors in seeds, seed extracts, seed flours and so on (e.g., tannins, saponins, cyanogenic glycoside).
Response: We have added information about antinutritional factors: “Unfortunately, seeds can also contain antinutrients that can have negative effects on human health. Saponins, tannins, phytates, lectins, or cyanogenic glycosides can be brough up as an example [20,21]. According to Pojić et al. (2014), the main antinutritional factors of hemp seed meal are phytates, glucosinolates, trypsin inhibitors, and condensed tannins [22]. Bueno-Borges et al. (2018) have reported, that sacha inchi seeds contained tannins, phytic acid, saponins, lectins, and trypsin inhibitors, though some of these compounds could be almost completely eliminated by thermal processing [21].”
L58-77 Please move this part in the section 2 “Chemical contents of seeds”. Please add information on different approaches used to extract bioactive compounds from seeds (from conventional to green extraction methods) and how the extraction parameters (T, solvent, time, other parameters) affect the bioactive compounds content and the biological activities of the extract, fraction, flour and so on. It is an important aspect.
Response: We have moved this paragraph to section 2 “chemical content of seeds”; we have added information on extraction methods and the effect of extraction parameters on the yield of compounds etc.: “There are many different methods that can be used to extract phytochemicals from plant materials. Some of the conventional methods include heat reflux extraction, maceration, Soxhlet extraction, infusion, percolation, or decoction. [51,52]. Apart from choosing the extraction method, the type of eluent must be carefully selected as well. It should be chosen in accordance with the hydrophilic or amphiphilic nature of the target compound, as well as other criteria [53]. Generally, methanol is better for extraction of polyphenols with low molecular weight, while aqueous acetone is more efficient in extraction of flavanols that have higher molecular weight. Anthocyanidins, in turn, can be obtained with weak organic acids, like acetic acid or formic acid [52]. Different combinations might be used to get specific results. For instance, the yield of phenolics extracted from berry seed meal was highest when a combination of methanol, acetone, and water (7:7:6, v/v/v) was used, while acetone-water (80:20 v/v), methanol-water, and water gave smaller amounts [6]. Naturally, the use of different eluents will result in different chemical content of the extract, which in turn will cause differences in biological activity. For example, the ethyl acetate fraction of sea buckthorn seed extract has been shown to possess significantly higher antioxidant activity than the n-hexane fraction [8]. Other criteria that influence the results of phytochemical extraction include temperature, extraction time, and purification procedures. Increasing the temperature and extraction time might lead to higher yield, because of improved solubility, and decreased surface tension and viscosity. However, many phytochemicals can become hydrolyzed in higher temperatures, so extra care must be taken to choose the optimal conditions. Purification is often needed to get rid of unwanted materials, like proteins, carbohydrates, or lipids. Different purification strategies include solid phase extraction (SPE), column chromatography, or liquid-liquid extraction [52]. In recent years, many modern green extraction methods have been developed as well. Green extraction methods use less or no organic solvents to minimize impact on the environment and reduce costs. Examples include microwave and ultrasound-assisted extraction, supercritical fluid extraction, pressurized liquid extraction, enzyme-assisted extraction, and pulsed electric field extraction [51,52].”
L78-89 Please better explain the choice of the eight seeds. There are other seeds with the same characteristics. Why authors selected the eight seeds? Which are the peculiar traits of these seeds? Are there major differences (chemical composition, health benefits, nutrient content) among these seeds? Please be more specific.
Response: We have added more information about the choice of seeds: “These eight species were chosen to represent different types of chemical compounds found in seeds and illustrate interesting biological activities that have the potential to be utilized in medicine and nutritional science.”
L115-118 Please better describe this part. The biological activities of these compounds have to be described with relevant references. Please describe the bioactive compound, its health benefits, and the related seed (source).
Response: We have described the most important groups of compounds, and provided examples of compounds, related seeds and health benefits: “Terpenoids can be divided into several groups, including triterpenoids and tetraterpenoids. Lycopene, a multi-functional compound present in Gac seeds can be brought up as an example [43–47]. Phytosterols, like campesterol or stigmasterol are natural steroids that can be found in many different plant species. Many studies have noted their positive effect on the cardiovascular system [48]. Tannins are a large group of polyphenols, that can be divided into hydrolyzable tannins and condensed tannins. Hydrolyzable tannins contain a central core of glucose or another polyol that can be esterified with gallic acid (gallotannins) or hexahydroxydiphenic acid (ellagitannins). For example, sanguiin H-6 is a hydrolyzable tannin abundant in blackberry seeds. It shows a wide array of biological activities, including antioxidant, antibacterial, antitumor, anti-angiogenic, estrogenic, and anti-osteoclastogenic [2,49–53]. Condensed tannins are oligomers or polymers of flavan-3-ol linked through an interflavan carbon bond. They are also called proanthocyanidins; examples include procyanidin B2 present in grape seeds [54]. Saponins are steroid and triterpenoid glycosides present in many different plant species. They display a wide range of biological properties, both beneficial and detrimental [55]. For example, saponins from Gac seeds had anti-inflammatory activity, but showed cytotoxic properties in mice [56,57].”
L122-125 Please be more specific. Which are the potential pharmaceutical applications of seed proteins and polysaccharides? Add some examples from literature
Response: We have added examples of biological activity and potential applications: “Proteins, protein hydrolysates, and peptides have shown a wide array of biological activities, including antioxidant, antimicrobial, anticancer, or antihypertensive. Bioactive peptides can be obtained from natural sources, like dairy, eggs, fish, nuts, legumes, cereals, fruits, or seeds. They can be formed during external hydrolyzation in the laboratory, in vivo digestion, or food processing. Polysaccharides are also an interesting area of research. They have been investigated for their antioxidant, immunomodulatory, antitumor, anticoagulant, and anticancer properties. Moreover, they can be modified by various methods (e.g. acetylation, selenization, or phosphorylation) to improve their effectiveness. Proteins and polysaccharides have showed various promising biological effects in animal models, but there is a need for more clinical trials that would allow them to be introduced as functional foods, nutraceuticals, and pharmaceuticals [39–45].”
L126-167 Please add the bioavailability and bioaccessibility of seed compounds with specific examples. Please be more specific.
Response: We have added specific examples: “For example, bioavailability of quercetin is 16-27.5%, luteolin 4.10-26%, kaempferol 2-20%, rutin 8.2%, apigenin 3-5%, and isorhamnetin 2.64% [72].”
“Bioavailability of phytosterols is quite poor. For example, only 1.9% of campesterol and 0.51% of sitosterol could be absorbed into the body [48].”
L155-167 Rewrite this part. It is confusing and not correct in English
Response: This part was rewritten: “Converting the doses used in animal studies to human equivalent doses (HED) can be done with the help of various methods. Two such methods are “no observed adverse effect level” (NOAEL) and “minimum anticipated biological effect” (MABEL). Dose adjustment must include different metabolic rates between large and small animals, as well as pharmacokinetics. After the initial calculation, the dose is further modified to minimalize the risk of adverse effects. For the purpose of this review, we will convert animal doses to HED. We will use a method based on the differences in body surface area (as in Nair and Jacob 2016), though it must be noted that this is a simplified approach, and if these compounds or preparations would be used in clinical trials, the doses would have to be adjusted further.”
Section 4. Chemical content of seeds from selected plant species and biological activity of seed extracts, fractions, oil, proteins, lipids, polysaccharides, and isolated compounds. Please re-organize the review’s structure. In particular two sections should be added: section 4 “Chemical content of seeds eaten with whole fruit and biological activity of seed extracts, fractions, oil, proteins, lipids, polysaccharides, and isolated compounds” and section 5 “Chemical content of seeds eaten separately and biological activity of seed extracts, fractions, oil, proteins, lipids, polysaccharides, and isolated compounds”. In addition, the order of presentation within each seed should be: 1 Chemical content of seed, 2 Biological activities of the whole extract, 3 Biological activities of fractions, isolated compounds, proteins, lipids, polysaccharides. Please follow the same order for all the eight seeds.
Response: We have reorganized the manuscript as advised.
L179-180 Please add the concentration found.
Response: concentration has been added - ellagic acid (653.81±66.84 μg/g) and sanguiin H6 (457–675 μg ellagic acid equivalents/g)
L193-195 Interesting result. Please explain the reason in the text.
Response: We have added more information about this: “The oxygen radical absorbing capacity (ORAC) assay indicated, that insoluble-bound phenolics had higher hydroxyl radical-scavenging abilities than free or esterified fractions. Insoluble-bound phenolics were also more efficient at preventing cupric ion-induced LDL cholesterol oxidation. This was caused by the gradual release of phenolics from their insoluble-bound forms, which is illustrated by the increase in activity at longer incubation times. Blackberry seed meal extracts had higher ferric-reducing power than the extracts from black raspberry and blueberry meals. Here, the activities of free, esterified, and insoluble-bound phenolics were similar, which underlines the need for different assays in antioxidant studies [6].”
L196-210 Authors should add the modulation of gut microbiota for other seed mentioned in the review (e.g., grape seeds doi.org/10.1186/s13099-022-00505-0, doi.org/10.1039/C7FO02028G, doi.org/10.1002/mnfr.201601082; moringa seed doi.org/10.1016/j.jff.2018.05.056; sea buckthorn doi.org/10.1039/C9FO01232J, doi.org/10.1016/j.ijbiomac.2017.09.090; gac doi.org/10.3390/ijms22052640 and so on). Revise
Response: We have added information about effect of the seeds on gut microbiota where possible (often, we found studies that involved extracts or powder from whole berries, pomace, or other parts of the fruit e.g. aril; in our review, we chose to focus only on the seeds):
Grape: “Commercial grape seed extract (GSE) facilitated the recovery of gut microbiota caused by antibiotic treatment in mice fed with high-fat diet. 200 mg/kg of GSE increased the abundance of Verrucomicrobia and decreased the amount of Actinobacteria. Moreover, it partially restored the content of Akkermansia in the feces and increased the relative abundance of Alloprevotella from 0.0075% to 0.0113%. Human equivalent dose of GSE would be approximately 16 mg/kg, which means that a 60 kg person would have to ingest 960 mg [110]. In another study, seed proanthocyanidins influenced the microbiota of female rats fed with a standard diet. A dose of 500 mg/kg decreased the number of Firmicutes and increased the number of Bacteroidetes and Proteobacteria. The abundance of Bacteroidaceae and Porphyromonadaceae was increased considerably. In contrast, the number of Ruminococcacea and Dehalobacteriaceae were decreased in comparison to control. In this case, HED is approximately 80 mg/kg (4800 mg for a 60 kg person) [111].”
Sea buckthorn: “Sea buckthorn seed oil could modulate the gut microbiota of hamsters fed with a high cholesterol diet. In experimental groups, 50 or 100% of lard was replaced with the oil. Both doses increased the abundance of Bacteroidales_S24-7_group, Ruminococcaceae, and Eubacteriaceae, at the same time decreasing the number of Firmicutes. As a result, the ratio of Firmicutes to Bacteroidetes was decreased [147]. In another study, Yuan et al. assessed the effect of sea buckthorn seed protein on the profile of intestinal microbes in streptozotocin-induced diabetic mice. The mice received 50,100, or 200 mg/kg of seed protein. Four-week treatment ameliorated streptozotocin-induced changes in the fecal content of Bifidobacterium, Lactobacillus, Bacteroides, and Clostridium coccoides [148].”
Hemp: “Ben Necib et al. (2022) studied the effect of hemp seeds on gut microbiota in mice fed with a high-fat, high sucrose diet. 15% of fat in the diet was substituted with fat from the seeds. This percentage of hemp seed fat in human diet could be attained by daily consumption of 37 g of ‘Finola’ hemp seeds, which is a reasonable amount. The abudance of Clostridiaceae 1 and Rikenellaceae was higher in the group supplemented with hemp seeds, but the Shannon diversity ratio and the proportion of Firmicutes to Bacteroidetes were not affected [165].”
Sacha inchi: “In a hyperuricemic rat model, supplementation of 0.42 g/kg BW of seed press-cake protein hydrolysate improved the diversity of gut microbiota. Moreover, it increased the abundance of probiotic bacteria like Akkermansia and Alistipes, at the same time reducing the Streptococcus population [180]. Sacha inchi oil had similar effects. A dose of 1.5 mL/kg reversed gut microbiota dysbiosis in rats fed with high-fat diet. The number of Roseburia, Turicibacter, and Butyrivibrio increased, while the levels of Enterobacteriaceae, Escherichia, and Bacteroides was reduced [182].”
L224-227 Please explain the intrinsic and extrinsic pathway in the text with relevant references
Response: We have added a short explanation: “Extrinsic pathway is considered to be the first step in plasma-mediated hemostasis, and is activated by tissue factor present in subendothelial tissue. Intrinsic pathway is a parallel pathway for thrombin activation, and begins with factor XII [57,61].”
L242-243 1140 mg per day? Please be more specific
Response: 1140 mg of blackberry seed polysaccharides twice a day
L246 replace consumption with dietary consumption. Moreover, is it calculated on a daily base?
Response: It has been replaced. The dose would have to be ingested twice a day.
L248-249 Please add a sub-section related to the antimicrobial activity of Rubus fruticosus L. seeds. The papers doi.org/10.3390/molecules26134057, doi.org/10.3390/molecules190810998, and doi.org/10.38001/ijlsb.1085539 are suggested for your analysis and discussion.
Response: a section about the antimicrobial activity of blackberry seeds has been added: “Non-polar extract from blackberry seeds (obtained by Soxhlet extraction) had antibacterial activity against Escherichia coli (IAL 2064) and Staphylococcus aureus (ATCC 13565). The growth of E. coli was inhibited by 99.4% - 33.4% (at concentrations of 3.3 - 0.42 μg/L), while the growth of S. aureus was inhibited by 90.7% and 33.3% (at concentrations of 33.3 and 1.67μg/L, respectively). Interestingly, extracts obtained by Bligh-Dyer and ultrasound extraction methods did not show antimicrobial effects [30].”
L249-277 Please add the anti-inflammatory activity of Rubus fruticosus L. seed compounds (see the review doi.org/10.3390/molecules190810998). Please use this paper for your analysis and discussion.
Response: We have added anti-inflammatory activity of other compounds present in the seeds: “Other compounds present in blackberry seeds also showed anti-inflammatory activity. For example, ellagitannins inhibited the transcription of NF-κB and decreased the secretion of IL-8 in an in vivo gastric inflammation model [74].”
L307-308 Which is the concentration of sanguiin H-6 in Rubus fruticosus L. seed? Is it practical for dietary consumption and associated health benefits? Otherwise, pharmaceutical products are more reasonable? Please explain as done for other seed compounds.
Response: concentration of sanguiin H-6 in the seeds has been added in the section “chemical content of blackberry seeds” (457–675 μg ellagic acid equivalents/g); in this study, sanguiin H-6 was administered topically, so we did not calculate HED.
L325-335 Please add more information on the antioxidant activity (see doi.org/10.1016/j.biortech.2007.08.063, doi.org/10.5010/JPB.2022.49.2.155)
Response: We have added the results from doi.org/10.5010/JPB.2022.49.2.155: “RC50 values of ethanol seed extract in DPPH and ABTS assays were 26.68 μg/mL and 39.30 μg/mL, respectively. The ferric reducing antioxidant power (FRAP) of the extract was 0.61 ± 0.01 mM FeSO4/mg [72]. “, but doi.org/10.1016/j.biortech.2007.08.063 was already described in the manuscript: “Black raspberry seed extract had antioxidant (DPPH•+ radical- and hydrogen peroxide-scavenging) activity (IC50 = 4.58 μg/mL), however it was lower than that of caffeic acid, quercetin, and (+)-catechin (IC50 = 1.76 μg/mL, 1.53 μg/mL, and 2.17 μg/mL, respectively). The ability to scavenge superoxide anions was more potent – the activity of extract made from seeds discarded during wine production was lower than that of quercetin but higher than (+)-catechin. The efficiency of Fe (III) chelation was comparable to that of tannic acid [68]”
L363-366 Rewrite this part. It is confusing and not correct in English
Response: The sentence has been rewritten: “Low molecular weight fraction and cyanidin-3 glucoside bound to murine norovirus-1 polymerase, inhibiting the expression of viral genes, which is likely the cause of their antiviral activity [72].”
L380-407 Please add the effect of the grape variety and the maceration time on the antioxidant activity of grape seed (see doi.org/10.3390/foods9101451 for example).
Response: it has been added” “Bosso et al. (2020) studied the effect of maceration time on antioxidant activity of the seeds from four different grape cultivars used in wine production. After short maceration (2 days) the ‘Uvalino’ variety had the highest ABTS, DPPH, and FRAP values, followed by ‘Grignolino’, ‘Nebbiolo’, and ‘Barbera’ varieties. After racking off, the highest values were recorded in the ‘Grignolino’ variety. Maceration decreased the antioxidant activity of all grape varieties. For example, the initial ABTS value for ‘Uvalino’ was 52.5 mmol TE/100 g DW, and after racking off it was 20.3 mmol TE/100 g DW [84].”
L458-467 Please merge this section with the section 4.3.4
Response: We have merged this part as advised
L468-528 Please short this part. Authors should report the main findings with a brief discussion, and with a particular focus on in vivo trials.
Response: We have shortened this section, especially the paragraph describing in vitro experiments
L558-561 Rewrite this sentence. It is confusing and not correct in English.
Response: The sentence has been rewritten: “Asbaghi et al. (2020) analyzed the results from 15 clinical trials that studied the effect of grape seed extract on glycemic control. At the dose of 300 mg a day or higher, GSE reduced the levels of fasting plasma glucose. On the other hand, it did not have any effect on the concentration of glycated hemoglobin A1c (HbA1c) [100].”
L561-562 Please add a new sub-section related to the antibacterial activity of grape seed extracts and whole grape pomace extracts including grape seeds. The different sensitivity of different pathogens as well as differences related to the grape cultivar should be included. The papers doi.org/10.1016/S0963- 9969(02)00116-3, doi.org/10.1016/j.foodcont.2011.09.016, doi.org/10.3390/molecules26195918, and doi.org/10.3390/molecules27051610 are suggested for your analysis and discussion.
Response: We have added information about antibacterial activity of grape seeds: “Two extracts from the seeds of ‘Bangalore’ variety had antibacterial activity against B. cereus, B. subtilis, S. aureus, B coagulans, E. coli, and P. aeruginosa. The minimal inhibitory concentration values (MIC) of the methanol:water:acetic extract were 900 ppm for B. cereus, B. subtilis, and B. coagulans, while the values for S. aureus, E.coli, and P. aeruginosa were 1000 ppm, 1250 ppm and 1500 ppm, respectively. MIC of the acetone:water:acetic extract for B. cereus, B. subtilis, and B. coagulans was 850 ppm, while in the rest of bacteria the inhibitory activity was the same in both extracts. The extracts were more effective against Gram-positive bacteria [107]. This is consistent with the results of other authors, who have noticed that extracts from grape pomace show weaker inhibition against Gram-negative bacteria. This phenomenon is attributed to the presence of polysaccharide cell wall, which can limit the penetration of polyphenols into the cell [108]. Grape seeds inhibited the growth of Helicobacter pylori as well. The MIC values of muscadine seed extract varied from 256 to 1024 μg/mL in different strains, though the skin extract was more effective in some of them (MIC = 256 - 512 μg/mL) [109].”
L572-586 Please add a possible mechanism of antioxidant activity of Moringa oleifera proteins and peptides with relevant references.
Response: a proposed mechanism has been added: “M. oleifera seeds contain seven kinds of hydrophobic amino acids, namely alanine, valine, methionine, isoleucine, leucine, tyrosine, and phenylalanine. As hydrophobic amino acid content is an important factor that contributes to the antioxidant activity of bioactive proteins, it has been proposed that they might be one of the reasons for the radical-scavenging activity of M. oleifera seed proteins”
L602 500 mg/kg…in this case which is the HED? Please include this information
Response: We have added the HED (40 mg/kg)
Section 4.4.5 Please enrich this section with other results. The papers doi.org/10.1016/j.ijbiomac.2017.10.070, DOI: 10.21010/ajtcam.v14i2.30, doi: 10.22034/APJCP.2016.17.11.4929 are suggested for your analysis and discussion.
Response: Results from these studies have been added: “Adebayo et al. (2017) assessed the effect of various seed extracts and fractions on MCF-7 (breast cancer) cells. Hexane and dichloromethane fractions of crude ethanolic extract inhibited cell growth (IC50 = 130 μg/mL and 26 μg/mL, respectively) in a dose dependent manner. Crude water extract inhibited proliferation as well, but its efficiency was lower (IC50 = 280 μg/mL) [109]. Nano-micelle of Moringa oleifera seed oil also showed cytotoxic activity toward cancer cells. It significantly reduced the viability of MCF-7 (IC50 = 86.5 μg/mL), HCT 116 (IC50 = 49.1 μg/mL), and Caco-2 (concentrations of 60-100 μg/mL induced cell death in approximately 50% cells) cell lines [110]. Anticancer activity of M. oleifera seeds was studied in vivo as well. Seed lectin inhibited the growth of Ehrlich ascites carcinoma (EAC) cells. The mice were injected with EAC cell intraperitoneally, and then treated with seed lectin injections once a day for five consecutive days. Cell growth was inhibited by 15.27% (at the dose of 2 mg/kg) and 55% (at the dose of 4 mg/kg) [111].”
Section 4.4.6 Please better describe the results of the cited references. In particular, the concentration considered should be highlighted. In addition, other papers could be included (see doi.org/10.1016/j.ijfoodmicro.2020.108770 for example).
Response: new information was added: “Lastly, the seed extract showed antibacterial activity against S. aureus (at the concentrations of 50-250 μg/mL) and, to a lesser degree, E. coli (at the concentrations of 100-250 μg/mL) [108,112]. Seed meal extract obtained with 200 W ultrasonification power inhibited the growth of S. aureus and E. coli as well (minimal inhibitory concentrations (MIC) = 3.13 and 6.25 mg/mL). It was also effective against S. typhimurium (MIC = 25 mg/mL) and B. cereus (MIC = 3.13 mg/mL) [113]. Seed lectin, in turn, could prevent the development of Bacillus sp. and S. marcescens biofilm. Even low concentrations (0.325, 0.65, and 1.3 μg/mL) were effective against S. marcescens. Lectin reduced the growth of Bacillus sp. at all tested concentrations (0.65 – 41.6 μg/mL), but 20.8 and 41.6 μg/mL had especially strong activity, comparable to the antibiotic rifampicin [114].”
L619-620 Please briefly introduce the main biological activities (in vivo) described in this section of the moringin.
Response: We have added a short introduction: “MIC-1 (moringin) is an isothiocyanate present in M. oleifera seeds. It has attracted attention due to its wide range of biological activities, including anti-inflammatory and pain-relieving. Researchers have also studied its therapeutic potential in neurodegenerative disorders [101,115,116].”
L624-628 Rewrite. This part is hardly understandable. For example, antinociceptive properties should be explained in the text.
Response: We have rewritten this part and added an explanation on nociception: ” . Nociceptive pain results from acute noxious stimuli (e.g. chemical stimulation, heat, or mechanical force). When noxious stimulus persists long enough, nociceptive pain can transform into inflammatory pain through the release of pro-inflammatory factors from nociceptive neurons [139]. MIC-1 activated and desensitized transient receptor potential type ankyrin ion channel (TRPA1) in transfected HEK-293 (human embryonic kidney) cells. TRPA1 is involved in transduction of signal in response to different stimuli, including cold, pungent compounds, airborne irritants, and cannabinoids. Its agonists are known to decrease the sensation of pain [135]. These properties were also studied in vivo. 2% moringin cream administered topically alleviated neuropathic pain in mice with multiple sclerosis. The cream also had an anti-inflammatory effect [140].”
L658-662 Rewrite this sentence. Short sentences are more appreciated by the Journal’s readers. Please use short sentences
Response: this fragment has been revised: “In the continuation of the above-mentioned in vivo study by Bao et al., anti-inflammatory properties of niazirin were revealed as well. It significantly decreased the concentration of plasma TNF-α and increased the levels of interleukin 10 (IL-10). IL-10 is an anti-inflammatory cytokine that reduces the release of inflammatory mediators, and subsequently inhibits migration and proliferation of inflammatory cells [115].”
L686 Please explain the mean of total antioxidant status. Are cell lines used in this study? Which cell lines? How total antioxidant status is determined? Revise
Response: this was an in vivo study on humans; we have added more details to improve clarity: “In a clinical trial by Vashishtha et al., supplementation of healthy volounteers with seed oil capsules (0.75 ml/day) caused an increase in serum total antioxidant status (determined with the ABTS method) by 46.13 μmol Trolox equivalent/L [129].”
L696 the required daily intake….
Response: The dose conversion method that we are using is not suitable for proteins with molecular weight > 100 kDa. Here, the authors did not specify the molecular mass of seed proteins used in the study, so we did not calculate HED in this case.
L741-774 Why no improvement was observed in high-cholesterol diet? Please explain in the text
Response: the text has been revised to avoid confusion: “Vasorelaxant response was decreased in rabbits fed with a high cholesterol diet, but it was brought back to the control level when the animals also received sea buckthorn oil”
L745-749 Please add other results. The papers doi.org/10.1016/j.foodchem.2004.07.009 and DOI: 10.5897/AJB11.4150 are suggested.
Response: other results has been added: Chloroform, acetone, and methanol seed extracts also showed antibacterial activity against L. monocytogenes and Y. enterocolitica, as well as B. cereus, B. coagulans, and B. subtilis. Methanol extract had the strongest effect, and Y. enterocolitica was the most resistant to the inhibitory activity of all three extracts. This resistance could be attributed to the presence of lipopolysaccharides in the membranes of Gram-negative bacteria [134]. Arora et al. (2012) studied the effect of the extracts from sea buckthorn leaves, seeds, and pomace on 17 different strains of Gram-positive and Gram-negative bacteria. Though the leaves showed the strongest antibacterial activity overall, the seeds were more effective than the pomace and strongly inhibited the growth of Bacillus cereus (inhibition zone = 17.7 mm), P. aeruginosa (15.3 mm), and Salmonella enterica (14.0 mm) [135].
L764-768 Is there a link with the chemical composition of ripe vs unripe/partially ripe seeds? Please explain in the text
Response: new information has been added: “These results seem to be correlated with the concentration of phytochemicals. Total phenolic content was higher in the seeds from ripe fruits (2.39 ± 0.26) than in the seeds from partially ripe fruits (1.63 ± 0.22). This was also the case with total flavonoid content (1.88 ± 0.10 vs. 1.57 ± 0.09).”
L782-783 Delete. It is not relevant in this section
Response: this fragment has been delated.
L784-790 Which are the health benefits associated to the neurogenic properties of the gac seed extracts? Please include this information in the text.
Response: an explanation has been added: “Small molecule NGF mimetics are investigated as potential therapeutic agents in the treatment of neurodegenerative disorders. Due to their size and structure, they could pass through the brain blood barrier and act similarly to nerve growth factor. Nerve growth factor can facilitate neuronal growth and repair, but stimulating its synthesis comes with a risk of serious adverse effects. For this reason, there is a need for new substances that act similarly to NGF, but have less side effects [138].”
L802-803 Authors stated that ethanolic extracts exerted anticancer activity. Then, they wrote that ethanol extracts did not have such an effect. Please check this sentence and revise.
Response: The authors of the study used two types of ethanol extracts – one prepared with 70% ethanol and one prepared with 100% ethanol. We have added this information to the text to avoid confusion
L804-812 Please move this part at the end of 4.6.5 section, and revise the title of the 4.6.5 section.
Response: the section has been moved and the title revised: ” Cytotoxic activity of Gac seed extracts toward healthy and cancer cells (in vitro and in vivo)”
L815-822 Delete. This information is not important in the context of this manuscript. Please refer only to the chemical composition of hemp seeds.
Response: It has been deleted.
L832-833 Authors should revise sentences like this throughout the manuscript. The correct sentence is:
“Hemp seed protein hydrolysate showed antioxidative activity in vitro. It had good DPPH•+ radical, hydroxyl radical….”.
Please revise the sentences like this throughout the manuscript. It is very important for the acceptance of the paper.
Response: the sentences have been revised.
L844-857 Please add information about the concentration of the extracts with anti-inflammatory activity.
Response: concentrations have been added: Cannabisin F (at the concentration of 15 μM) and grossamide (at the concentration of 20 μM)
L847-850 Please report these results in one sentence. Thanks
Response: this fragment has been adjusted: “Lignanamides from hemp seeds had anti-inflammatory properties in vitro [142,145,146]. Cannabisin F (at the concentration of 15 μM) and grossamide (at the concentration of 20 μM) ameliorated neuroinflammation in BV2 murine microglia cell line stimulated by LPS. NF-κB activation was reduced, and the expression of IL-6 and TNF-α was inhibited [142,146]”
Section 4.7.5 The paper doi.org/10.1016/j.clinbiochem.2021.04.008 is suggested for your analysis and discussion in this section.
Response: We have added new information based on this paper: “In a study by Kaur et al. (2021) hempseed lipid fractions (10% of feed) ameliorated hypercholesterolemia in rats fed with a high-fat diet. HFD-induced renal damage was reduced as well. These effects have been linked to antioxidant and anti-inflammatory properties of hemp seeds. Furthermore, treatment with hempseed lipid fractions had similar efficiency to treatment with statins, which is a promising result that could lead to the development of new strategies for improving the condition of hypercholesterolemic patients. For these reasons, hemp seeds should be thoroughly investigated to further our understanding of their activity [148].”
L885-887 Rewrite this sentence. It is not correct in English
Response: The sentence has been rewritten: “Moreover, the content of CBD in the seeds is lower than in inflorescences and leaves. In ‘Finola’ variety the concentration of CBD in the seeds was 14.80±0.32 mg/kg. Leaves had slightly more CBD (16.54±1.20 mg/kg), but the highest concentration was recorded in female plants’ inflorescences (135.41±8.64 mg/kg).”
L902-903 Please enrich this section with the results of the cited references at L903. This section is too short. Revise
Response: results from two studies have been added: “Seed extract had radical-scavenging capacity as well. It could reduce the ABTS•+ cation (0.49 μmol TE/g seed). Interestingly, roasting the seeds improved their antioxidant activity against DPPH•+ (an 13% increase) and peroxyl radicals (an 22% increase) [158]. Sacha inchi oil, which is rich in alpha-linoleic acid, improved oxidative stress parameters in the liver of rats. Inclusion of the oil in the diet (10% feed) raised the con-centration of hepatic GSH and increased the activity of SOD, CAT, GPX, and glutathione reductase (GR) [159].”
L906-917 Rewrite this section. It is very confusing. Please report the main findings related to the immunomodulatory activity of sacha inchi seed polysaccharides and proteins. Revise
Response: The section has been rewritten “Apart from its antioxidant activity, PVLP-1 could also improve proliferation and viability of RAW 264.7 cells. It also upregulated the secretion of TNF-α, IL-1β, IL-6, and nitric oxide [162]. Albumin fraction of sacha inchi seed protein had similar effects as PVLP-1. It stimulated the proliferation of mouse spleen lymphocytes. The proliferation rates of cells treated with 10 and 20 μg/mL of albumins were higher than positive control (cells stimulated with LPS or Concanavalin A). The albumins increased levels of TNF- α as well [163].”
L927 Please use the MDPI style for this reference.
Response: Reference style has been corrected.
L930-931 Please add a section related to the anti-inflammatory activity of sacha inchi seeds/oils. The papers DOI: 10.31989/ffhd.v11i3.778, Quinteros, M., Vilcacundo, R., Carpio, C., & Carrillo, W. (2016). Digestibility and anti-inflammatory activity in vitro of sacha inchi (Plukenetia volubilis L.) proteins. Asian J. Pharm. Clin. Res, 9(3), 303-306, and doi: 10.3389/conf.fimmu.2015.05.00349 are suggested.
Response: the section has been added: “Quinteros et al. (2016) have reported, that sacha inchi proteins had anti-inflammatory activity. At the concentrations of 100-1000 μg/ml, they inhibited albumin denaturation by 7.77±0.3% - 78.13±0.44%. At the highest concentration, their activity was comparable to that of diclofenac sodium (200 μg/mL), which is an anti-inflammatory drug [170]. Anti-inflammatory activity of sacha inchi was studied in vivo as well. In rats with hyperuricemia, administration of 0.42 g/kg of seed press-cake protein hydrolysate decreased the expression of pro-inflammatory genes (Tlr4, Map3k8, Ikbke, Nlrp3, Pik3cg, Pik3ap1, and Akt3) in the kidneys [171]. Moreover, in a clinical trial by Alayón et al. (2019), sacha inchi oil reduced the levels of postprandial inflammation induced by eating a high-fat breakfast. 15 ml of seed oil ingested with the meal, decreased the concentration of IL-6 in the serum of metabolically healthy participants [172].”
L932-933 Please divide the Table 1 in two tables: the first one related to the active compounds and their biological activity of seeds eaten with the fruit, the second related to the the active compounds and their biological activity of seeds eaten separately. Revise
Response: We have divided and reorganized the table
L942-974 Delete general parts from the conclusion section. Please refer only to the biological activity of selected seeds, the chemical compounds associated with health benefits, the issues related to their efficacy in in vitro vs in vivo trials. The concentration usually found bioactive in vivo is not compatible with diet consumption of seeds. Therefore, the use of seed extracts for the preparation of pharmaceutical products is more realistic. Please add this information in the conclusion section
Response: the entire conclusions section has been revised. We have deleted general information, added a summary of biological activity of all seeds and compounds described in the text, and added the information about the need for pharmaceutical products when concentration of the active compounds in the seeds is insufficient:
“Blackberry seed extracts showed antioxidant, antimicrobial, and antigenotoxic activity. Seed flour could change the profile of gut microbiota, and polysaccharides modulated coagulation. Ellagic acid and sanguiin H-6 are prominent constituents of the seeds. Ellagic acid has well-documented anti-inflammatory activity, while sanguiin H-6 displays a wide array of biological activities, including antioxidant, antibacterial, antitumor, anti-angiogenic, estrogenic, and anti-osteoclastogenic.
Extracts from black raspberry seeds displayed antioxidant and antiviral activity. Polyphenols isolated from the seeds (gallic acid and cyanidin-3-glucoside) also inhibited viral infection, while seed oil had hepatoprotective properties in vitro.
Grape seed extracts had antioxidant, anti-inflammatory, antitumor, cardioprotective, hypoglycemic, and antibacterial activity. Moreover, they inhibited the oligomerization of amyloid β, modulated lipid metabolism, and facilitated the recovery of gut microbiota after treatment with antibiotics. Seed oil had antioxidant, anti-inflammatory, and hypoglycemic properties as well. Various compounds isolated from grape seeds (including proanthocyanidins and procyanidins) showed antioxidant, anti-apoptotic, antitumor, and cardioprotective activities, and modulated gut microbiota.
Extracts from the seeds of Moringa oleifera had antioxidant, anti-inflammatory, neuroprotective, cytotoxic, and antibacterial effects. Moringa seed oil showed anti-inflammatory, neuroprotective, and cytotoxic activities as well. Seed proteins were effective antioxidants, while seed lectin inhibited the growth of EAC carcinoma cells and had antibacterial properties. MIC-1 and niazirin had promising biological effects as well. MIC-1 had anti-inflammatory, antioxidant, pain-relieving, and antitumor activity. Niazirin showed hypoglycemic, antioxidant, anti-inflammatory, and anti-hypercholesterolemic effects.
Sea buckthorn seed extracts had antioxidant, hypoglycemic, anti-inflammatory, anti-hyperlipidemic, anti-hypercholesterolemic, and antibacterial effects. Moreover, seed oil showed antioxidant, antihypertensive, anti-hypercholesterolemic, and anti-hyperlipidemic activity. Seed proteins had anti-inflammatory, anti-hypercholesterolemic, and anti-hyperlipidemic properties, while flavonoids could reduce oxidative stress and inflammation. Both seed protein and oil could modulate gut microbiota as well. Total flavones from sea buckthorn seeds showed antihypertensive activity, while alkaloids protected embryonic rat cardiac cells from doxorubicin-induced toxicity.
Gac seed extracts had antioxidant, neurotrophic, antitumor, and cytotoxic effects. MCoCI (a chymotrypsin inhibitor) isolated from Gac seeds had antioxidant activity as well. Lignans and saponins had anti-inflammatory properties.
Hemp seeds and seed lipid fractions could decrease the levels of cholesterol in rats fed with high-fat diet. Seed proteins and polysaccharides showed antioxidant activity, while lignanamides reduced inflammation and had antitumor effects. Cannabidiol, a well-known constituent of hemp has been shown to possess many beneficial properties, including anti-seizure, cardioprotective, antioxidant, anti-inflammatory, antidepressant, and anticancer. Moreover, hemp seeds altered the content of of Clostridiaceae 1 and Rikenellaceae in the gut.
Sacha inchi seed extract had antioxidant activity. The seeds were also cytotoxic toward healthy human cells, but this could be attenuated through roasting. Seed oil had antioxidant activity as well. It also showed anti-hypercholesterolemic, antihyperlipidemic, anti-hypertensive, anti-inflammatory effects. Seed polysaccharide (PVLP-1) and protein has immunomodulatory activity. Moreover, proteins reduced inflammation and modulated gut microbiota, while polysaccharide had antioxidant properties.
Despite the promising results obtained in in vitro experiments we still do not have enough results from animal studies and clinical trials. Among the factors that determine in vivo effectiveness of extracts or chemical substances, bioavailability is of great importance. Some phytochemicals, like anthocyanins, are known for their poor bioavailability. This can pose a problem that would have to be resolved if we were to consider implementing these substances in medical scenarios [28,292]. Fortunately, there are ways to improve bioavailability of a compound – for example by converting it into smaller molecules, combining it with other compounds, or enclosing it into lipid vesicles, polymeric nanoparticles or other types of nanomaterials [46,51–53]. Assessing the safety of studied extracts, oils, or compounds is immensely important as well [3,68,152].
In the future, more emphasis should be put on animal studies and clinical trials. This step is needed to further our knowledge of the preparations and compounds that have well documented in vitro activity, and check if they can be used as functional foods, nutraceuticals, or pharmaceuticals. It must be noted, that oftentimes the concentration of active compounds in seeds is too low to be effective only through dietary consumption. When this is the case, using the extracts, fractions, oil, proteins, polysaccharides, or compounds obtained from seeds in pharmaceutical products is a more realistic option.”
L979-1650 References. Please revise the references, re-numbering the literature throughout the manuscript and adding the references suggested in the peer-review report. Thanks
Response: references have been revised.
Round 2
Reviewer 2 Report
Authors revised the original submission according to reviewer's comments. The review is improved and well organized. Some minor changes are suggested below:
L1157 How albumin denaturation is linked with anti-inflammatory activity? Please explain in the text.
L1083-1091 Please enrich this part with other results from literature
Conclusion section The conclusion section has been revised. However, it is too long. Please short to one third the conclusion section, reporting the main findings of the review and future perspectives.
Author Response
Authors revised the original submission according to reviewer's comments. The review is improved and well organized. Some minor changes are suggested below:
Thank you for reviewing the manuscript and providing such helpful comments. All of them have been taken into consideration when revising the manuscript.
L1157 How albumin denaturation is linked with anti-inflammatory activity? Please explain in the text.
Response: We have added an explanation: “In vivo protein denaturation is one of the causes of inflammation, and can lead to arthritis and other inflammatory diseases. For this reason, denaturation-preventing agents are valued as potential candidates for anti-inflammatory drugs [203].”
L1083-1091 Please enrich this part with other results from literature
Response: We have added other results: “Mokhtari et al. (2022) studied the effect of hemp seed oil on hyperlipidemic mice. The doses of 3.5 and 7 mg/kg decreased plasma total cholesterol by 45.7% and 57%, respectively. The concentration of triglycerides was decreased by 82% (at the dose of 3.5 mg/kg) and 87% (at the dose of 7 mg/kg). The levels of LDL-C were decreased by 61%, and the levels of HDL-C increased by 324%. These changes resulted in reduced atherogenic index and LDL/HDL ratio [187]. On the other hand, in a study by Majewski and JurgoÅ„ski supplementation of obese male Zucker rats with hemp seed oil (4% diet) decreased the concentration of HDL and triglycerides, but did not affect total cholesterol levels [188].”; Fig. 2 has been updated to include these results.
Conclusion section The conclusion section has been revised. However, it is too long. Please short to one third the conclusion section, reporting the main findings of the review and future perspectives.
Response: We have shortened the conclusion.